# HIERARCHY PRUNING FOR UNSEEN DOMAIN DISCOVERY IN PREDICTIVE HEALTHCARE

## ABSTRACT

Healthcare providers often divide patient populations into cohorts based on shared clinical factors, such as medical history, to deliver personalized healthcare services. This idea has also been adopted in clinical prediction models, where it presents a vital challenge: capturing both global and cohort-specific patterns while enabling model generalization to unseen domains. Since cohort boundaries naturally correspond to domain boundaries, addressing this challenge falls under the scope of domain generalization (DG), especially when domain labels are not explicitly available in EHR data. However, regular DG approaches often struggle in clinical settings due to the absence of domain labels and the inherent gap in medical knowledge. Moreover, the rich hierarchical structures embedded in medical ontologies have rarely been explored as a basis for deriving clinically meaningful domain partitions. Hence, we propose UDONCARE, a hierarchy-guided method that iteratively divides patients into latent domains and decomposes domain-invariant (label) information from patient data. On two public datasets, MIMIC-III and MIMIC-IV, UDONCARE shows superiority over eight baselines across four clinical prediction tasks with substantial domain gaps, highlighting the potential of medical knowledge in guiding clinical DG problems.

## 1 INTRODUCTION

The digitization of clinical data, notably electronic health records (EHR), has transformed healthcare by enabling efficient computational analysis. Current deep learning techniques have also achieved significant gains in diagnosis, mortality, and readmission prediction tasks (Poulain & Beheshti, 2024; Jiang et al., 2024). Still, these models trained on the training (source) data often suffer performance drops when applied to the test (target) data under domain shifts, that is, distributional changes across patient groups, such as data from different hospitals (Perone et al., 2019; Koh et al., 2021). Consequently, handling domain shifts is a prerequisite for alleviating performance degradation in clinical predictive models (Yang et al., 2023a; Wu et al., 2023). It also aligns with the objective of most domain generalization (DG) methods, such as meta-learning (Balaji et al., 2018; Dou et al., 2019), adversarial learning (Ganin et al., 2016; Li et al., 2018b), and latent-domain techniques (Matsuura & Harada, 2020; Wu et al., 2023).

In this work, we focus on tackling DG problem in clinical settings, whereas most recent models have been developed for image classification. However, directly transferring these regular DG methods will encounter two clinical-specific obstacles: (1) Domain IDs, which are naturally defined in image datasets (e.g., dog & cat), are unseen in most EHR datasets, but most DG solutions require the presence of domain IDs (Wu et al., 2023). Some studies treat each patient as unique domain (Dou et al., 2019; Yang et al., 2023a), which is overly fine-grained and unstable. Others rely on broader categorizations (e.g. institute & admission period), which overlook clinical heterogeneity (Zhang et al., 2021a; Guo et al., 2022). (2) Even though some DG methods do not rely on domain IDs (Arjovsky et al., 2019; Liu et al., 2021b), they overlook clinical semantics. For instance, Matsuura & Harada (2020); Wu et al. (2023) cluster patient features to form latent domains, but the resulting partitions are highly sensitive to training data. In practice, patient groups can vary significantly, even over longer admission periods, since they reflect only feature-level similarity without capturing the progression of medical concepts. Hence, it is crucial to construct robust domains with explicit definitions grounded in clinical relevance.

To address these challenges, we explore the following research question: *Instead of assuming the presence of domain IDs, can we leverage medical knowledge to guide models in discovering domains that are both adaptive and clinically meaningful?* In most hospitals, visiting patients are treated based on their medical history, which is expressed through medical concepts shown in their admissions. For instance, when dealing with heart failure patients, hospitals may categorize heart failure as a distinct domain or group it with other cardiovascular diseases. Similarly, in medical ontologies like ICD-9-CM, heart failure corresponds to a leaf node under a higher-level node grouping cardiovascular disease, and such hierarchical relation motivates us to use a pruning algorithm to identify appropriate ancestor nodes for domain partitioning.

However, translating medical ontologies into a form suitable for DG is not a simple problem. Most DG methods discover domains by clustering patients in the feature space (Matsuura & Harada, 2020; Wu et al., 2023), which makes it difficult to incorporate ontology structures for categorization. Moreover, existing clinical prediction studies often use ontologies to enrich patient representations by using graph neural networks (Jiang et al., 2024), rather than to guide how patient cohorts should be partitioned. Thus, a key gap remains: medical ontologies have rarely been used as structural rules for discovering domains. To bridge this gap, we aim to leverage ontologies not as auxiliary features, but as principled rules that shape domain formation in a clinically meaningful way.

To this end, we propose UDONCARE, a framework that leverages medical ontologies to discover latent domains at multiple levels of abstraction. It ensures the discovered domains remain consistent with clinical semantics while maintaining flexibility for adaptive generalization. Specifically, a pruning algorithm is developed to merges similar concepts on hierarchies and generate soft labels as domain IDs for patients. To explicitly remove domain features from patients, we leverage a mutual learning network, which learns domain-invariant (label) representations upon the orthogonal factorization. Finally, domain assignments and feature extraction are updated jointly through an iterative collaborative inference mechanism, allowing the pruning module to adapt domain categorization according to input data and task settings. Our main contributions are enumerated as follows:

- To the best of our knowledge, this is the first work using medical ontologies to tackle clinical DG problems. It reveals the potential of medical ontologies in finding latent domains for handling covariates, rather than serving as feature enrichment (Lu et al., 2021; Jiang et al., 2024).

- UDONCARE shows accurate prediction across four vital predictive tasks on two public datasets, outperforming both clinical DG baselines (Yao et al., 2022; Wu et al., 2023) and regular DG baselines. UDONCARE boosts the AUPRC score by $5 - 20\%$ over the best baselines.

- We conduct detailed analyses to show that UDONCARE addresses domain shifts through accurate domain partitioning and invariant feature learning, without sacrificing computational overhead.

## 2 PRELIMINARY

**DG Problems on EHR Data.** Single patient data $\mathbf{x}^{(i)}$ consists of a longitudinal sequence of admissions $\{V_1^{(i)}, V_2^{(i)}, \ldots, V_T^{(i)}\}$. We denote the entire set of medical concepts as $\mathcal{C} = \{c_1, c_2, ..., c_{|\mathcal{C}|}\}$, and each admission contains a subset of $\mathcal{C}$. Most predictive models predict clinical outcomes $\mathbf{y}^{(i)} \in \{0, 1\}^d$ at a future visit $V_{t+1}$, and they develop a feature extractor $f_{\phi,k}(\cdot) : \mathbf{x}_k \mapsto \mathbf{p}_k$ to encode historical data $\mathbf{x}_k$ into patient-level embedding $\mathbf{p}_k$ before getting downstream predictions:

$$\mathbf{p} = \mathbf{p}_1 \oplus \cdots \oplus \mathbf{p}_k \oplus \cdots \oplus \mathbf{p}_K, \text{ where } \mathbf{p}_k = f_{\phi,k}(\mathbf{x}_k) \in \mathbb{R}^h. \tag{1}$$

where $f_{\phi,k}$ is the encoder for feature key $k$, $\oplus$ denotes vector concatenation across $K$ feature keys, and $h$ is the embedding dimension. When $f_\phi(\cdot)$ are trained on source training data sampled from distribution $P_s$, we often expect the learned model to also perform well on target data drawn from $P_t$ by assuming $P_s = P_t$. However, it is possible that these two distributions differs a lot $P_s \neq P_t$ due to spatial and temporal shifts, which motivates the need for DG solutions to factorize label information that remains invariant across unseen clinical environments (i.e. domains).

**Concept-Specific Hierarchy.** In EHR data, certain medical concept $c_i \in \mathcal{C}$ always originates from a hierarchical encoding system, such as ICD-9 (Organization et al., 1988) and ATC (Nahler & Nahler, 2009) codes. We define a concept-specific hierarchy $\mathcal{H}$ of $H$ levels, and denote $n_i^{(h)}$ as the $i$-th node on level $h$. Leaf nodes at level $H$ represent actual codes via the mapping $m : c_i \mapsto n_i^{(H)}$ with node

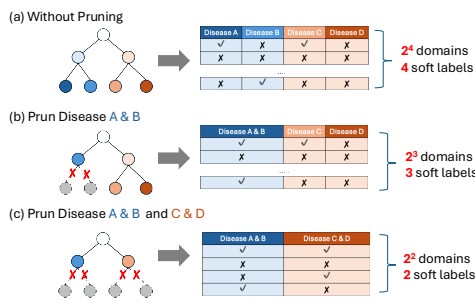

Figure 1: **The Overall Framework of UDONCARE.** The forward structure adds a domain pathway for mutual learning, extending beyond the backbone pathway of conventional predictive models. During training, we first feed patient data **x** into the backbone pathway, which learns patient features **p** through $f_{\phi,k}(\cdot)$ and produces the output prediction $\hat{y}_p$. In parallel, we obtain $\hat{y}_h$ from invariant features **h** along the domain pathway by applying DiscoveryAlgo$(\cdot)$, $g_{\theta,k}(\cdot)$, and $h(\cdot)$. Here we iteratively adapt latent domains in **M** and update parameters on both pathways by ground truths $y$.

feature $\mathbf{e}_i$ stored in $f_{\phi,k}$, and $\mathrm{Desc}(\cdot)$ denotes the set of descendant nodes. Note that, the root node $n_1^{(1)}$ at the top level subsumes all nodes in $\mathcal{H}$, and any two leaf nodes $n_i^{(H)}$ and $n_j^{(H)}$ share at least one common ancestor. In this work, we only focus on the disease hierarchy (i.e. ICD-9-CM), as incorporating treatments and drugs yields marginal gains (see Appendix G). However, UDONCARE can be extended to incorporate additional feature keys or ontologies if needed.

## 3 METHODOLOGY

UDONCARE iteratively operates two main steps (as shown in Figure 1): (1) developing a pruning algorithm for medical hierarchies to discover latent domains; (2) learning invariant (label) information by factorizing patient features in projection space. A notation table is also provided in Appendix C.

### 3.1 STEP 1: HIERARCHY-GUIDED DOMAIN DISCOVERY

While domain IDs are unobserved in EHR datasets, it is intuitive that patients with similar medical histories (concepts) often belong to the same domain. We can divide patient cohorts by treating the multi-hot vector $\mathbf{v}_t$ from admissions as a soft domain label. However, the number of latent domains grows exponentially with the larger vocabulary size $|\mathcal{C}|$ (i.e. $2^{|\mathcal{C}|}$). Here, we design a hierarchy-guided domain discovery algorithm that assigns and updates domain IDs for patients' admissions. Our goal is to prune overly fine-grained nodes, thereby forming a smaller set of ancestors that still covers all concepts, as shown in Figure 2. An assignment matrix **M** can be generated to query domain IDs via

Figure 2: A simple illustration of hierarchy-guided domain discovery.

$$\mathbf{M} \;:=\; \mathrm{DiscoveryAlgo}\Big(\{\mathbf{x}^{(i)}\}_{i=1}^{N_{\mathrm{tr}}}\Big) \;\in\; \{0,1\}^{N_{\mathrm{tr}} \times |\mathcal{C}'|}, \tag{2}$$

where $N_{\mathrm{tr}}$ is the number of training patients, $|\mathcal{C}'|$ is the pruned vocabulary size of medical concepts. It merges fine-grained codes into fewer, higher-level clusters, allowing patients with or without a particular disease to occupy different latent domains as needed.

**Initialization on Domain IDs.** Following the previous settings (Lu et al., 2021; Jiang et al., 2024), we only focus on patients with multiple admission, where there are $T \geq 2$ records. This sequence should be then converted into a single vector that consolidates all prior visits. Hence, we aggregate

patient data $\{\mathbf{x}_1, \mathbf{x}_2, \ldots, \mathbf{x}_T\}$ into a unified representation $\mathbf{X} = \bigvee_{t=1}^{T} \mathbf{x}_t$, where medical concepts shown in each admission are merged to form a comprehensive medical history. Note that, $\mathbf{X}$ is the most fine-grained domain assignment, which initializes $\text{DiscoveryAlgo}(\cdot)$.

**Initialization on Node Features.** After getting $\mathbf{X}$ for each patient, concept-specific ontology $\mathcal{H}$ with node features are required to decide whether fine-grained medical concepts group diseases into higher-level clusters or are preserved. We initialize leaf-node features $\mathbf{e}_i \in \mathbb{R}^h$ by: (1) for **present code** $c_i$ in $\mathcal{S}$, $\mathbf{e}_i$ is initialized from embedding layer $\text{E}(e_1, \ldots, e_{|\mathcal{C}|})$ in $f_{\phi,k}(\cdot)$; (2) for **absent code** $c_i$ in $\mathcal{S}$, $\mathbf{e}_i$ is its embedding of entity name through ClinicalBERT (Huang et al., 2019). The feature of a parent node $\mathbf{e}_{n_i}^{(h-1)}$ is then computed from the embeddings of its descendants at level $h$:

$$\mathbf{e}_i^{(h-1)} := \frac{1}{|\text{Desc}(n_i^{(h-1)})|} \sum_{n \in \text{Desc}(n_i)} \mathbf{e}_n^{(h)}, \tag{3}$$

which extends $\text{E}(e_1, \ldots, e_{|\mathcal{C}|})$ to $\text{E}(e_1, \ldots, e_{|\mathcal{H}|})$ over the entire hierarchy. Still, it fails to capture the hierarchical distances in $\mathcal{H}$. For example, two leaf nodes might have similar features without sharing the same parent node. Under this scenario, their lowest common ancestor (LCA) $n_{\text{LCA}}(d_i, d_j)$ could be an alternative reflecting such similarity. Hence, we mimic the principle of hierarchical clustering (Johnson, 1967) in this stage. Concretely, we determine the most similar node pair $(d_i, d_j)$ based on their cosine similarity $\cos(\mathbf{e}_{d_i}, \mathbf{e}_{d_j})$, and then update the embedding of their LCA

$$\mathbf{e}_{\text{LCA}(d_i, d_j)} \leftarrow \text{Average}\big(\mathbf{e}_{\text{LCA}(d_i, d_j)}, \mathbf{e}_{d_i}, \mathbf{e}_{d_j}\big). \tag{4}$$

This process continues sequentially down the similarity list until the remaining highest similarity falls below the threshold $\rho = 0.3$. See Appendix I for more details about the information flow.

**Node Scoring.** For each node $n \in \mathcal{H}$, we define $S(n)$ to identify which node is a good "candidate" for final selection. Motivated by the information gains (Song & Ying, 2015), $S(n)$ involves three indicators (see in Appendix J), coverage $\text{cov}(n)$, purity $\text{pur}(n)$, and depth $\text{dep}(n)$, via

$$S(n) = \alpha \cdot \exp(\text{pur}(n)) + (1 - \alpha)\big(\text{cov}(n) \times \text{dep}(n)\big)$$
$$= \alpha \cdot \exp\big(\mathbb{E}_{m \in \mathcal{M}}[\text{sim}(\mathbf{e}_n, \mathbf{e}_m)]\big) + (1 - \alpha)\big(\frac{|\mathcal{M}|}{|\mathcal{L}|} \cdot \frac{h}{H}\big) \tag{5}$$

where $\mathcal{M}$ is equivalent to $\text{Desc}(n)$; $\mathbb{E}(\cdot)$ denotes the mathematical expectation; $\alpha$ and $\exp(\cdot)$ act as scaling factors, which regularize the selection avoiding either too low or high level. Consequently, the score matrix $S(s_1, \ldots, s_{|\mathcal{H}|})$ is obtained after scoring all nodes in the hierarchy.

**Hierarchy Pruning.** Once $S(n)$ is computed, we perform a bottom-up pass over $\mathcal{H}$ to generate a candidate set of pruned nodes. Let $p$ be the parent node with children $\{c_1, \ldots, c_r\}$. There are three possible situations upon comparing score $S(p)$ with its children scores $\{S(c_i)\}^r$:

- If $S(p) > \max(\{S(c_i)\}^r)$, we include parent node $p$ and exclude its children.
- If $S(p) < \min(\{S(c_i)\}^r)$, we discard parent node $p$ and select all children.
- Otherwise, we tentatively include $p$ but mark it for further resolution in the next step.

Note that, if $p$ is discarded in the above cases, $\{c_1, \ldots, c_r\}$ are promoted to become the children of $p$'s parent. After the first iteration on the score matrix, a candidate subset $\mathcal{C}_0$ will be generated. However, such a result can be considered as a suboptimal solution, since marked candidates still require further evaluation. Here a list of tuples $A$ of length $N$ is adopted to trade off each flagged parent-child pair $A[n] := (p, \{c_1, \ldots, c_r\})$ to find an optimal result via a chosen search strategy.

**Domain Searching.** Given $N$ flagged pairs, the complexity of searching approaches grows exponentially $O(2^N)$. Here we adjust a Beam-Search algorithm (Lowerre, 1976) that gets decisions within the top-k pruning candidates $O(kN)$. For each flagged pair, we either (1) unify them by including the parent or (2) retain the children as distinct pruned nodes. Note that those pairs that have not yet been decided default to retaining parent node to ensure complete coverage. We adopt the Silhouette Score (Shahapure & Nicholas, 2020) as the evaluation metric. Specifically, UDONCARE fist apply UMAP (McInnes et al., 2018) on leaf node features for dimensionality reduction, and it then calculate the score on this full hierarchical partition, reflecting the global separation quality under the current pruning scheme. This process iterates over all flagged node pairs in order by updating pruning subsets in $\{\mathcal{C}_1, \mathcal{C}_2, \ldots, \mathcal{C}_N\}$ to make selection with the optimal evaluation result.

**Domain Decision.** To this end, we obtain an updated vocabulary $\mathcal{C}' \subseteq \mathcal{H}$ of selected higher-level nodes and update the domain-assignment matrix $\mathbf{M}$ by linking each patient's admission records (originally from $\mathcal{C}$) to these pruned clusters in $\mathcal{C}'$, where $|\mathcal{C}'| \leq |\mathcal{C}|$. If patient $p^{(i)}$ has at least one leaf code $d$ that descends from the pruned node $p_j \in \mathcal{C}'$, we update $\mathbf{M}[i,j] = 1$; otherwise, $\mathbf{M}[i,j] = 0$. Note that, the output matrix $\mathbf{M}$ define domain categorization in terms of node features.

## 3.2 STEP 2. MUTUAL FORWARD LEARNING

Each domain can be viewed as a latent representation $\mathbf{r}$ sampled from a meta domain distribution $p(\cdot)$, so that we can identify $\mathbf{r}$ and then factorize $p(y|\mathbf{x})$ into $\int p(y|\mathbf{x}, \mathbf{r})p(\mathbf{r}|\mathbf{x})\, d\mathbf{r}$ by approximating $q(\mathbf{r}) \sim p(\mathbf{r}|\mathbf{x})$ given data samples $\mathbf{x}$. Subsequently, a domain encoder $p(\mathbf{r}|\mathbf{x})$ and a label predictor $p(y|\mathbf{x}, \mathbf{r})$ are needed for inference. Here we parameterize the domain encoder $p(\mathbf{r}|\mathbf{x})$ as a network $g_\theta(\cdot)$ with parameter $\theta$. Since the pruned output matrix $\mathbf{M}$ (see Section 3.1) maps each training sample $\mathbf{x}$ to $\mathbf{m}$ (soft-label domain IDs), we apply $g_\theta(\cdot)$ to $\mathbf{m}$ to estimate the domain factor $\mathbf{r} := g_\theta(\mathbf{m})$. Although $\mathbf{r}$ represents a probabilistic domain variable, we implement $g_\theta$ as a deterministic Multi-Layer Perceptron (MLP) for the prediction task. Next, we compute invariant features using a non-parametric function $h(\cdot) : (\mathbf{r}, \mathbf{p}) \mapsto \mathbf{h}$, which fuses $\mathbf{r}$ (domain-level representation) and $\mathbf{p}$ (patient-level representation) as input features for the label predictor $p(y|\mathbf{x}, \mathbf{r})$.

**Self-Supervised Domain Encoder.** The main concerns on training domain encoder is how to ensure $g_\theta(\cdot)$ can extract valid domain information from patients, which is ignored by some works (Finn et al., 2017; Li et al., 2018a; Yang et al., 2023a). A regulation method is then developed during the encoder training phase. Concretely, pseudo domain labels $\mathbf{m}$ help us divide patients into latent domains, where averaging patient-specific features $\bar{\mathbf{p}}$ could provide guidance for $g_\theta(\cdot)$ in identifying domain information. Hence, we adopt a pretraining task and update $\theta$ based on patient embeddings $\mathbf{p}$ from $f_\phi(\cdot)$ by minimizing loss function

$$\mathcal{L}_r[g_\theta(\mathbf{m}), \bar{\mathbf{p}}] := \mathrm{MSE}(\mathbf{r},\ \mathbb{E}[\mathbf{p}|\mathbf{m}]) + \frac{\|\mathbf{r}_\mu - \mathbf{p}_\mu\|_{\mathcal{F}}^2}{\|\mathbf{p}_\mu\|_{\mathcal{F}}^2}. \tag{6}$$

where $\mathbb{E}[\mathbf{p}|\mathbf{m}]$ denotes the average embedding associated with domain IDs, and $\|\mathbf{r}_\mu - \mathbf{p}_\mu\|_{\mathcal{F}}^2$ measures the Maximum Mean Discrepancy (Borgwardt et al., 2006) with the norm $\mathcal{F}$ to reduce distributional gaps. The subscript $\mu$ indicates batch-level averages. The domain encoder $g_\theta(\cdot)$ can then approximate domain features $\mathbf{r}$ through both patient-level inputs $\mathbf{p}$ and $\mathbf{m}$.

**Invariant Feature Projection Learning.** In equation 6, both $\mathbf{r}$ and $\mathbf{p}$ are rescaled into a shared vector space with comparable magnitudes. Hence, we can directly apply an orthogonal projection approach (as in early studies (Bousmalis et al., 2016; Shen et al., 2022; Yang et al., 2023a)) to obtain the invariant feature $\mathbf{h}$ by subtracting the parallel component of $\mathbf{p}$ in this shared vector space. We formalize this in $h(\cdot)$ as shown in equation 7:

$$\mathbf{h} := \mathbf{p} - \tilde{\mathbf{r}},\ \text{where } \tilde{\mathbf{r}} = \mathbf{r} \cdot \langle \frac{\mathbf{p}}{\|\mathbf{r}\|}, \frac{\mathbf{r}}{\|\mathbf{r}\|} \rangle. \tag{7}$$

Here, $\tilde{\mathbf{r}}$ is the component of $\mathbf{p}$ that is parallel to $\mathbf{r}$ with domain covariates, while $\mathbf{h}$ is the remainder and thus invariant to domain shifts. We thus obtain invariant features $\mathbf{h}$ without additional parameters, and $h(\cdot)$ serves as an essential pre-processing step before making prediction.

## 3.3 TRAINING AND INFERENCE

**Iterative Training.** To train UDONCARE, we feed each data sample $\mathbf{x}$ into the hierarchy-pruning module to obtain its latent domain $\mathbf{m}$, and then perform two cross-reference steps under a mutual learning architecture. Rather than updating the model continuously in each epoch, we adopt an iterative training strategy, which prior studies (Cui et al., 2019; Sofiiuk et al., 2022) have shown can reduce training time while maintaining comparable predictive performance.[1] We iteratively update the model weights and regenerate domain assignments every 20 epochs in our experiment. Before each iteration, we reinitialize the parameters in $g_\theta(\cdot)$, because the input shape of $\mathbf{m}$ may change due to updated code-level embeddings. We also provide the pseudo-code of UDONCARE in Appendix B.

---

[1]For example, we set iterations $I = 3$ and epochs $N = 100$ by first obtaining pretrained parameters for 40 epochs and then updating $\mathbf{M}$ iteratively every 20 epochs, yielding a total of 100 epochs.

Table 1: Statistics of MIMIC-III and MIMIC-IV datasets.

| Dataset | # patients | Max. # visit | Avg. # visit | Avg. #$\mathcal{D}$/visit | Avg. #$\mathcal{P}$/visit | Avg. #$\mathcal{M}$/visit |
|---|---|---|---|---|---|---|
| MIMIC-III | 6,497 | 42 | 2.66 | 13.06 | 4.54 | 33.71 |
| MIMIC-IV | 49,558 | 55 | 3.66 | 13.38 | 4.70 | 43.89 |

**Mutual Inference.** After the orthogonal projection, we apply the network $q_\xi(\cdot)$ (operates on space $\mathbf{p}$) as a post-step to parameterize the label predictor $p(y|\mathbf{x}, \mathbf{r})$; It is also available to parameterize $p(y|\mathbf{x})$ through the regular decoder network $d_\eta(\cdot)$ fed by patient embeddings from backbone $f_\phi(\cdot)$.

$$p(y|\mathbf{x}) \sim \hat{y}_p = d_\eta(\mathbf{p}) = d_\eta(f_\phi(\mathbf{x}))$$
$$p(y|\mathbf{x}, \mathbf{r}) \sim \hat{y}_h = q_\xi(\mathbf{h}) = q_\xi(h(g_\theta(\mathbf{m}), \mathbf{p})) \tag{8}$$

Both $q_\xi(\cdot)$ and $d_\eta(\cdot)$ are linear classifiers with learnable parameter matrices. A loss function $\mathcal{L}$ is then applied to add label supervision for downstream predictive tasks. We integrate these two predictors into a collaborative framework, with the mutual inference objective:

$$\mathcal{L}_p = \mathbb{E}_{(\mathbf{x},y)\sim\mathcal{S}_{train}}\ell(\hat{y}_p, y) + \lambda \cdot D_{\mathrm{KL}}(\hat{y}_p || \tilde{y})$$
$$\mathcal{L}_h = \mathbb{E}_{(\mathbf{x},y)\sim\mathcal{S}_{train}}\ell(\hat{y}_h, y) + \lambda \cdot D_{\mathrm{KL}}(\hat{y}_h || \tilde{y}) \tag{9}$$

where $D_{\mathrm{KL}}(\hat{y}_* \| \tilde{y})$ denotes the KL Divergence (Van Erven & Harremos, 2014) with the same $\lambda$ value in $\mathcal{L}_p$ and $\mathcal{L}_h$. Within each loss, $\ell(\cdot)$ denotes the binary cross entropy and $\tilde{y}$ is the average probability of $\hat{y}_p$ and $\hat{y}_h$. These two losses are calculated jointly $\mathcal{L}_p + \mathcal{L}_h$ to let $d_\eta$ and $q_\xi$ regularize one another, stabilizing the learning of $q_\xi$ with less parameters. Following the DG setting, UDONCARE applies $\hat{y}_h$ as the final prediction.

## 4 EXPERIMENTS

### 4.1 EXPERIMENT SETUPS

**Predictive Tasks.** We evaluate our approach on four representative tasks: (1) **Mortality Prediction**, which determines whether a patient will pass away by a specified time horizon after discharge. This is a binary classification task. (2) **Readmission Prediction**, which checks if a patient will be readmitted within a predefined window (e.g., next 15 days) following discharge. This is also framed as a binary classification. (3) **Diagnosis Prediction**, which forecasts the set of diagnoses (ICD-9-CM codes) for the patient's next hospital visit based on prior visits. This requires multi-label classification. (4) **Drug Recommendation**, which suggests a set of medications (ATC-4 codes (Nahler & Nahler, 2009)) for the upcoming visit, also formulated as multi-label classification. These tasks reflect diverse clinical needs and provide a rigorous benchmark for evaluating DG methods.

**Evaluation Metrics.** Both readmission and mortality prediction are binary classification tasks, we calculate the Area Under the Precision-Recall Curve (AUPRC) and the Area Under the Receiver Operating Characteristic Curve (AUROC) scores due to the imbalanced label distribution. For the drug recommendation task, we evaluate predictions of all DG approaches by AUPRC and F1-score, following the same setting as ManyDG (Yang et al., 2023a) (i.e. $d < 120$). For the diagnosis prediction tasks, we decide accurate prediction by weighted $F_1$ score as in Timeline and top-10 recall as in DoctorAI (Choi et al., 2016), since the former one measures the overall prediction on all classes (i.e. $d > 4500$) and the latter one have concentration on positive code with low frequency.

**Datasets & Data Split.** We conduct experiments on two publicly available EHR databases, **MIMIC-III** and **MIMIC-IV**, which are widely used in clinical prediction (Johnson et al., 2016; 2023). MIMIC-III covers ICU admissions from 2001 to 2012, while MIMIC-IV spans 2008 to 2019. To avoid overlapping time ranges with MIMIC-III, we only retain patients from the years 2013–2019 in MIMIC-IV. For each set of experiments, we extract 6,497 and 49,558 patients with multiple visits ($T \geq 2$) from both datasets as shown in Table 1. Different from random data splitting, we evaluate our model's performance across temporal gaps, following the approach in SLDG (Wu et al., 2023). We define a temporal grid based on the year of each patient's most recent visit. Specifically, patients in MIMIC-IV (MIMIC-III) whose last visit occurred after 2017 (2010) are assigned to the target test set, while those with earlier visits are used as the source training/validation set. We

Table 2: **Performance comparison of four prediction tasks on MIMIC-III/MIMIC-IV.** We report the average performance (%) and the standard deviation (in bracket) over 5 runs.

| Model | Task 1: Mortality Prediction | | | | Task 2: Readmission Prediction | | | |
|---|---|---|---|---|---|---|---|---|
| | MIMIC-III | | MIMIC-IV | | MIMIC-III | | MIMIC-IV | |
| | AUPRC | AUROC | AUPRC | AUROC | AUPRC | AUROC | AUPRC | AUROC |
| Oracle | 16.73 (0.51) | 70.35 (0.55) | 8.46 (0.53) | 68.92 (0.47) | 73.42 (0.47) | 69.74 (0.51) | 67.37 (0.12) | 66.89 (0.11) |
| Base | 11.31 (0.62) | 55.21 (0.95) | 3.97 (0.47) | 59.13 (0.76) | 50.13 (0.88) | 45.27 (0.71) | 48.34 (0.24) | 45.72 (0.31) |
| DANN | 12.54 (0.55) | 63.08 (0.72) | 4.41 (0.45) | 63.82 (0.46) | 58.29 (0.63) | 49.22 (0.78) | 53.13 (0.11) | 47.91 (0.23) |
| CondAdv | 13.75 (0.49) | 65.53 (0.57) | 5.65 (0.62) | 64.27 (0.68) | 61.31 (0.45) | 52.45 (0.51) | 56.95 (0.14) | 50.77 (0.19) |
| MLDG | 13.12 (0.47) | 64.48 (0.61) | 4.75 (0.38) | 62.75 (0.57) | 60.12 (0.53) | 51.03 (0.62) | 55.62 (0.23) | 49.57 (0.28) |
| IRM | 13.74 (0.45) | 65.21 (0.58) | 4.14 (0.52) | 62.36 (0.61) | 60.97 (0.47) | 52.02 (0.58) | 56.40 (0.14) | 50.58 (0.17) |
| PCL | 13.52 (0.52) | 64.79 (0.58) | 5.35 (0.49) | 64.70 (0.55) | 60.47 (0.46) | 51.56 (0.55) | 56.08 (0.28) | 50.99 (0.26) |
| ManyDG | 14.24 (0.51) | 65.98 (0.55) | 6.06 (0.31) | 64.66 (0.32) | 62.38 (0.42) | 53.19 (0.54) | 57.81 (0.25) | 52.34 (0.24) |
| SLDG | 13.07 (0.50) | 63.89 (0.60) | 4.58 (0.44) | 63.24 (0.60) | 59.78 (0.49) | 50.81 (0.53) | 56.92 (0.14) | 52.85 (0.16) |
| UDONCARE | **15.82 (0.33)** | **69.04 (0.42)** | **6.81 (0.27)** | **66.73 (0.48)** | **71.17 (0.35)** | **67.28 (0.39)** | **61.61 (0.10)** | **58.62 (0.25)** |

| Model | Task 3: Drug Recommendation | | | | Task 4: Diagnosis Prediction | | | |
|---|---|---|---|---|---|---|---|---|
| | MIMIC-III | | MIMIC-IV | | MIMIC-III | | MIMIC-IV | |
| | AUPRC | F1-score | AUPRC | F1-score | w-$F_1$ | R@10 | w-$F_1$ | R@10 |
| Oracle | 80.25 (0.12) | 67.23 (0.31) | 74.31 (0.25) | 61.28 (0.22) | 26.73 (0.12) | 39.22 (0.18) | 28.12 (0.11) | 40.53 (0.16) |
| Base | 68.54 (0.13) | 47.65 (0.32) | 66.94 (0.18) | 53.13 (0.18) | 21.51 (0.14) | 30.83 (0.20) | 20.07 (0.12) | 31.52 (0.18) |
| DANN | 75.32 (0.21) | 60.82 (0.34) | 69.63 (0.27) | 53.43 (0.26) | 21.84 (0.13) | 34.51 (0.22) | 24.05 (0.14) | 35.21 (0.20) |
| CondAdv | 76.81 (0.19) | 64.18 (0.28) | 71.48 (0.15) | 55.62 (0.29) | 22.81 (0.11) | 36.48 (0.20) | 26.13 (0.12) | 37.35 (0.18) |
| MLDG | 74.92 (0.22) | 59.13 (0.31) | 70.29 (0.27) | 56.77 (0.16) | 21.54 (0.15) | 33.93 (0.21) | 24.17 (0.13) | 34.72 (0.19) |
| IRM | 69.23 (0.19) | 62.47 (0.33) | 69.12 (0.14) | 54.57 (0.18) | 22.41 (0.14) | 33.07 (0.22) | 23.54 (0.15) | 34.12 (0.21) |
| ManyDG | 77.04 (0.20) | 63.94 (0.30) | 71.26 (0.19) | 55.27 (0.19) | 23.12 (0.12) | 36.17 (0.21) | 25.91 (0.13) | 37.04 (0.20) |
| UDONCARE | **78.31 (0.18)** | **66.42 (0.32)** | **73.07 (0.33)** | **59.23 (0.14)** | **24.79 (0.10)** | **38.05 (0.19)** | **27.31 (0.11)** | **39.41 (0.17)** |

also divide the dataset into training, validation, and test subsets using a fixed ratio of 75%:10%:15%. Note that, to avoid information leakage in clinical modeling, we ensure that no patient's earlier visits appear in the training set if their later visits are included in the test set.

**Baselines.** We compare UDONCARE with two naïve baselines, five general DG baselines and two most recent clinical DG baselines: (1) Naïve Baselines: **Oracle**, trained backbone encoder directly on the target data, and **Base**, trained solely on the source data. The difference between metrics from Oracle (upper bound) and Base (lower bound) can show the distribution gaps between source and target data. (2) Typical DG Baselines: **DANN** (Ganin et al., 2016), **CondAdv** (Isola et al., 2017), **MLDG** (Li et al., 2018a), **IRM** (Arjovsky et al., 2019), and **PCL** (Yao et al., 2022). (3) Clinical DG Baselines: **ManyDG** (Yang et al., 2023a), and **SLDG** (Wu et al., 2023). Since drug recommendation and diagnosis prediction are multi-label classification, we drop PCL and SLDG in these two tasks due to their setting limitation. Note that, all baselines follow the same source/target definition as in the Data Split section, and the test set remain unseen to all models except Oracle. The results reported in following sections are the performance evaluated on the test set. More details of the baselines can be found in Appendix D, and UDONCARE is implemented as described in Appendix E.

## 4.2 MAIN RESULTS

Table 2 presents results on four classification tasks using MIMIC-III and MIMIC-IV. First, the performance gap between the Oracle and Base methods is substantial, showing the presence of considerable domain differences. Focusing on mortality prediction in MIMIC-III, DANN (Ganin et al., 2016) and MLDG (Li et al., 2018a), both relying on coarse domain partitions—show minimal improvements, likely due to difficulties in extracting consistent features from coarse partitions. PCL (Yao et al., 2022) exhibits a slight gain through proxy-to-sample relationships. Meanwhile, IRM (Arjovsky et al., 2019) and CondAdv (Isola et al., 2017) perform better by incorporating regularization or recurrent structures for binary temporal event prediction. Among clinical-specific baselines, ManyDG (Yang et al., 2023a) achieves the best results by leveraging mutual reconstruction, and SLDG (Wu et al., 2023) sees only modest improvement due to its reliance on the most recent admissions. Notably, UDONCARE surpasses all baselines across all tasks. Specifically, UDONCARE boosts the AUPRC score by around 5% for mortality in MIMIC-III, 8% for mortality in MIMIC-IV, and around 21% and 19% for readmission in MIMIC-III and MIMIC-IV, respectively. We also extend experiments by using GAMENet (Shang et al., 2019b) and CGL (Lu et al., 2021) as the backbone, and evaluate model generalization across hospitals on eICU dataset (see Appendix F).

Table 3: Cosine similarity of linear weights on $\mathbf{p}$, $\widetilde{\mathbf{r}}$, and $\mathbf{h}$

| Cosine Similarity | Mortality Prediction | Readmission Prediction | Drug Recommendation | Diagnosis Prediction |
|---|---|---|---|---|
| $W_d(\mathbf{p} \to \text{labels})$ vs. $W_d(\mathbf{h} \to \text{labels})$ | $0.7831 \pm 0.0174$ | $0.4813 \pm 0.0391$ | $0.6546 \pm 0.0237$ | $0.8654 \pm 0.0198$ |
| $W_d(\mathbf{p} \to \text{domains})$ vs. $W_d(\widetilde{\mathbf{r}} \to \text{domains})$ | $0.3427 \pm 0.0088$ | $0.1957 \pm 0.0314$ | $0.2753 \pm 0.0274$ | $0.2133 \pm 0.0036$ |
| $W_d(\mathbf{p} \to \text{labels})$ vs. $W_d(\mathbf{p} \to \text{domains})$ | $0.1239 \pm 0.0126$ | $0.0794 \pm 0.0392$ | $0.1251 \pm 0.0212$ | $0.0972 \pm 0.0095$ |

∗ $W_{\text{cls}}(\cdot)$ represents the learned linear weights. As an illustration, $W_{\text{cls}}(\mathbf{p} \to \text{domains})$ denotes training a linear model on $\mathbf{p}$ to predict domain IDs, after which the weights are extracted. Cosine similarity scores are averaged across all classes and evaluated over 3 runs.

### 4.3 MORE QUANTITATIVE ANALYSIS

**Effectiveness of Decomposition.** Following the setting of Shen et al. (2022); Yang et al. (2023a), a linear classifier $d$ can be trained on the embedding to predict either (i) labels or (ii) domains. After training such a predictor, the cosine similarity can be calculated in terms of the learned weights to quantify the feature dimension overlaps. Note that $\mathbf{p}$, $\widetilde{\mathbf{r}}$, and $\mathbf{h}$ are normalized dimension-wise to ensure that each dimension is comparable. The results are shown in Table 3. In general, the third row has lower cosine similarities than the first two rows, which indicates that there is mostly non-overlap between feature dimensions predicting labels and domains. Moreover, the first two rows give relatively higher similarity and imply the domain and label information are separated from $\mathbf{p}$ into domain features $\widetilde{\mathbf{r}}$ (scaling from $\mathbf{r}$) and invariant features $\mathbf{h}$. It provides the quantitative evidence that UDONCARE stores domain and label information along distinct dimensions. In addition, We provide more explanations in Appendix L. Under the same setting as Table 3, we also attach the convergence process in detail (see Appendix H).

**Ablation Analysis.** We evaluate whether the designed domain-discovery algorithm is effective for prediction. Given a lookup embedding table for condition concepts, we need to group similar codes to reduce dimensionality. Hence, (a) k-Means clustering, (b) hierarchical clustering, and (c) tree pruning of the information gain algorithms can be adopted to simplify the process. We use drug recommendation tasks as an example to assess performance. The results can be found in Figure 3. The left figure illustrates the trade-off between accuracy and computing time at three iterations. We observe that k-Means performs the worst, largely because of its limitations in determining the optimal number of clus-

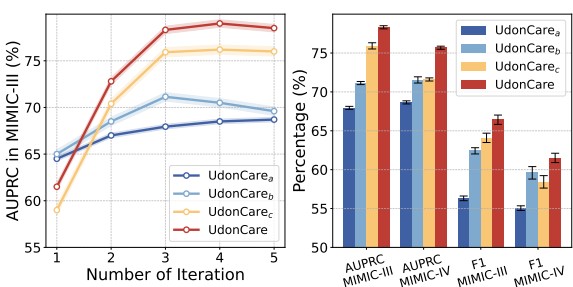

Figure 3: **Effectiveness of Domain Discovery.** The left figure shows the effect of the number of iteration on AUPRC on MIMIC-III dataset, and the right one shows comparison among variants upon UDONCARE.

ters via grid search. Hierarchical clustering performs better than k-Means but lacks structured guidance from the medical hierarchy. Tree pruning outperforms both methods by leveraging medical ontologies, demonstrating the importance of knowledge-driven clustering. Moreover, UDONCARE outperforms all these methods by incorporating more precise domain IDs through iterative beam-search updates. These results highlight the critical role of medical ontologies in domain discovery and the advantages of adaptive refinement for learning meaningful structures.

**Runtime Analysis** Lastly, we compare the training time of UDONCARE with two other clinical DG baselines, ManyDG and SLDG, as shown in Table 4. All runtimes are measured on an NVIDIA L40S GPU. We find that an iterative training strategy effectively balances computational overhead and performance for the entire framework. We observe that UDON-CARE requires a training time comparable to SLDG, since both rely on iterative parameter updates that reduce the frequency

Table 4: **Running Time Comparison of Drug Recommendation (seconds per epoch).** Note that SLDG only use the most recent admissions for prediction.

| Model | MIMIC-III | MIMIC-IV |
|---|---|---|
| Base | $3.206 \pm 0.1219$ | $6.943 \pm 0.2342$ |
| ManyDG | $5.462 \pm 0.2648$ | $9.215 \pm 0.3781$ |
| SLDG | $4.518 \pm 0.0256$ | $8.439 \pm 0.1329$ |
| UDONCARE | $4.320 \pm 0.1473$ | $7.871 \pm 0.2415$ |

of model adjustments during inference. ManyDG consumes more time than others, primarily because its domain assumption spawns numerous latent domains for subsequent computations.

**Effect of Training Data Size.** Next, inspired by Yang et al. (2023a); Jiang et al. (2024), we investigate how the volume of training data impacts model performance by conducting a comprehensive experiment in which the training set size ranges from 1% to 100%. Such a comparison is meaningful for examining how well models generalize with few domain samples. We evaluate drug recommendation on MIMIC-IV, since its complexity poses a challenging setting for prediction under varying EHR data sizes. All reported metrics are averaged over five independent runs. The results in Figure 4 indicate that all models show reduced performance in both AUPRC and F1-score when

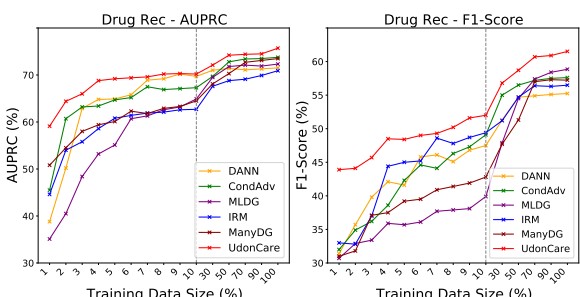

Figure 4: **Performance by Training Size.** We evaluate drug recommendation on MIMIC-IV, and values on the x-axis indicate % of the entire training data. The dotted lines divide two ranges: [1, 10] and [10, 100].

labeled data are scarce, particularly below 10% of the training set. However, UDONCARE maintains a considerable edge over other baselines, suggesting that its co-training strategies effectively minimize information loss even when domain features are limited. Notably, MLDG lacks a certain level of resilience against data limitation, likely due to unique domain assignment, which might not work for the situation that existing patients with only few admissions (less than 3) on EHR datasets.

## 4.4 CASE STUDY OF DISCOVERED DOMAIN

To illustrate how UDONCARE behaves, We evaluate discovered domains on diagnosis prediction using MIMIC-III, Here, we focus on three kinds of heart related conditions from diseases of the circulatory system: heart failure (HF, 15 types), essential hypertension (EH. 3 types), and acute rheumatic fever (ARF, 7 types). In the left table of Figure 5, we can observe that the mid-level "heart failure", high-level "hypertensive disease", and several leaf-level nodes are chosen to represent these subtypes. These decisions reflect the same patterns of code-embedding geometry (right scatterplot plot): HF codes form a compact cluster, EH codes cluster even more tightly and lie near the HF group, while ARF codes are widely dispersed. Moreover, it is also consistent with medical knowledge: HF and EH are clinically cohesive, while ARF shows greater heterogeneity due to variable cardiac involvement. Thus, these results align with both data-driven and clinical logic.

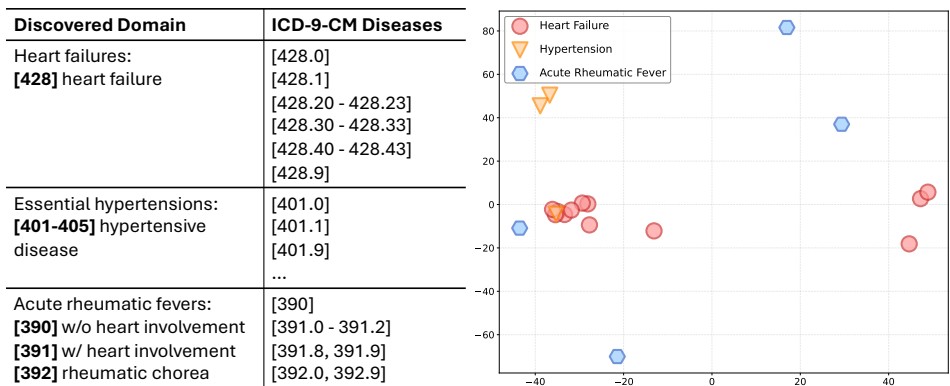

Figure 5: **Domains among diseases of the circulatory system identified by UDONCARE.** The left table shows how the discovered domains correspond to specific ICD-9-CM diseases (codes) in the hierarchy, while the right figure is the scatter plot using t-SNE upon code-level embeddings.

## 5 RELATED WORK

**Domain Generalization (DG).** DG is pursuing adjusted models which are specially designed to remove domain-covariate features from hidden representation (Muandet et al., 2013). A significant amount of work has been dedicated to solve performance drops on target domain across diverse scenarios like computer vision (Zhou et al., 2022; Ding et al., 2022), and they can be generally categorized as three different ways: (i) An intuitive way is to minimize the empirical source risk, either domain alignment (Ganin et al., 2016; Li et al., 2018b; Zhao et al., 2020) and invariant learning technique (Liu et al., 2021a; Zhang et al., 2022a; Wang et al., 2022) aim to convey little domain characteristics to acquire task-specific features. (ii) Contrastive learning (Kim et al., 2021; Jeon et al., 2021; Yao et al., 2022) becomes an alternative for data augmentation, studies employed the contrastive loss function to reduce the gap of representation distribution in one category. (iii) Meta-learning (Balaji et al., 2018; Dou et al., 2019) and ensemble-learning (Cha et al., 2021; Chu et al., 2022) approaches handle domain shifts through dynamic loss functions. However, they typically predefine either numerous or few domains in clinical settings (Wu et al., 2023), which motivates us to design a precise and efficient way to discover latent domains from learnable parameters.

**DG in Clinical Prediction.** Empirical evidence (Perone et al., 2019; Koh et al., 2021) has shown that EHR predictive models often suffer performance drops when transferred to new records with different data distributions. To address this, most existing works (Zhao et al., 2020; Zhang et al., 2021b) consider domain adaptation to handle potential domain shifts across multiple hospitals (Reps et al., 2022; Zhang et al., 2022b) and different time periods (Guo et al., 2022). Recently, a growing number of studies (Guo et al., 2022; Hai et al., 2024) consider model generalization by mitigating the patient-specific domain shifts, providing a more flexible alternative in more scenarios. For instance, Yang et al. (2023a) learns invariant features by treating each patient as a unique domain; Wu et al. (2023) develops a mixture-of-domain method to divide patients into latent domains by features of medical concepts. However, they primarily address domain categorization with simplifying heuristics such as linear dependencies (Li et al., 2020), while the potential of incorporating medical knowledge beyond EHR data remains unexplored in clinical DG problems.

An additional discussion about ontology-based predictive models can be found in Appendix A.

## 6 CONCLUSION

This paper develops UDONCARE, a novel framework for unseen domain discovery in predictive healthcare. Under the guidance of medical ontologies, our method discovers and iteratively adapts domain categorization. Extensive evaluations on two MIMIC datasets demonstrate that UDONCARE outperforms state-of-the-art baselines across multiple tasks. For example, in mortality prediction, UDONCARE surpasses other baselines by 4-8% in AUPRC; On readmission tasks, it also gains up to 6%, while drug recommendation has 5-10% improvements. Despite these gains, UDONCARE remains comparable computational efficiency to other clinical DG baselines. These results demonstrate the good model generalization with knowledge-driven domain discovery in clinical practices. Future work can further reveal more benefits of medical knowledge in robust clinical predictions. Beyond the structured, code-based EHR setting studied in this work, the ontology-guided design of UDONCARE suggests broader applicability to multi-modal clinical prediction and more general ontology-rich problems. We extend the discussion of these potential directions in Appendix K.

## ETHICS STATEMENT

We acknowledge that we have read and adhered to the ICLR Code of Ethics in the preparation and presentation of this work. In line with the principles of responsible stewardship, we are committed to upholding high standards of scientific excellence, honesty, and transparency in our research. We have conducted and presented this work with integrity, giving proper acknowledgment to the contributions of others and ensuring that our findings are reported accurately and reproducibly. We recognize the importance of minimizing potential harms and have reflected on the broader societal impacts of our research, including implications for human well-being and the natural environment. Consistent with the values of fairness and inclusivity, we support the equitable participation of all individuals in research and seek to promote accessibility and inclusiveness in both our methods and outcomes. We further respect the privacy and confidentiality of data that inform scientific discovery, and we endeavor to ensure that our work contributes positively to society, advances knowledge responsibly, and aligns with the long-term public good. At present, we do not identify any specific ethical concerns associated with this research.

## REPRODUCIBILITY STATEMENT

We have taken several steps to ensure the reproducibility of our work. Details of the proposed method (Section 3.1 and Section 3.2) with training & inference procedures (Section 3.3) are provided, with pseudo code (Appendix B), implementation details (Appendix E) and complete results (Section 4.2 and Section 4.3) included. Furthermore, we supply anonymized source code of UDON-CARE in the supplementary materials to facilitate replication of our experiments.

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

APPENDIX

## A  RELATED WORK: HIERARCHY-AWARE PREDICTIVE MODELING

EHR prediction models can be broadly categorized into three types: RNN/CNN-based (Choi et al., 2017; Tong et al., 2025a), GNN-based (Lu et al., 2021; Jiang et al., 2024), and Transformer-based models (Shang et al., 2019a; Poulain & Beheshti, 2024; Tong et al., 2025b). Concretely, most graph-based studies (Choi et al., 2017; Shang et al., 2019a; Lu et al., 2021) utilize hierarchical medical classifications like ICD-9 (Organization et al., 1988) and ATC (Nahler & Nahler, 2009) to determine medical concept similarity by assuming diseases closer in the hierarchy share more characteristics, as reflected in similar embeddings. However, this method can be biased as it typically fails to capture complex relationships beyond simple parent-child links, such as complications or comorbidity, leading to sub-optimal predictions (Xu et al., 2023; 2024; Hu et al., 2024). Moreover, there is few research integrating these hierarchical structures with DG techniques, which could enhance model robustness across diverse healthcare settings. Properly leveraging hierarchical relationships in DG could improve the domain discovery process, ensuring models account for variance in disease manifestation across different patient demographics and regional practices. Thus, integrating hierarchy-aware modeling with DG approaches holds potential for developing more accurate and personalized predictive models in EHR, catering to the nuanced needs of global healthcare environments.

## B  PSEUDO CODE FOR UDONCARE

Since the training and inference phase has been explained in the main paper, we conduct our pseudo code by two consecutive phases:

---

**Algorithm 1** Overview of UDONCARE

---

**Input:** EHR dataset $\mathcal{S}$ with patient's data $\{\mathbf{x}^{(i)}\}_{i=1}^{N_{\mathrm{tr}}}$; Feature extractor $f_\phi$ with defined backbone (e.g. Transformer).

1: *// When iteration $I = 3$ and epochs $N = 100$*
2: **for** epoch $\in \{1, 2, \ldots, 40\}$ **do**
3:     *// Backbone Pathway*
4:         Decode $\hat{y}_p \leftarrow d_\eta(f_{\phi,k}(x))$ with Equation 8;
5: **end for**
6: Obtain learned $\phi$; Initialize the hierarchy $\mathcal{H}$;
7: **for** iteration $\in \{1, 2, 3\}$ **do**
8:     *// Hierarchy-Guided Domain Discovery*
9:         Define & Update look-up table $\mathbf{M}$ via Step 1-4;
10:         Assign domain IDs $\mathcal{M} : \mathbf{x} \mapsto \mathbf{m}$ with Equation 2;
11:     *// Self-Supervised Domain Encoding*
12:         Initialize $g_\theta$; Pretrain $g_\theta$ by minimizing Equation 6;
13:         **for** epoch $\in \{1, 2, \ldots, 20\}$ **do**
14:             *// Domain Pathway*
15:             Obtain patient embeddings $\mathbf{p}$ with Equation 1;
16:             Get domain features $\mathbf{r}$ by $g_\theta(\mathbf{m})$;
17:             Decompose invariant features $\mathbf{h}$ with Equation 7;
18:             Decode $\hat{y}_p, \hat{y}_h$ with Equation 8;
19:             Minimize co-training loss $\mathcal{L}_p, \mathcal{L}_h$ with Equation 9;
20:         **end for**
21: **end for**

**Output:** Trained models $f_\phi$, $g_\theta$, $d_\eta$, $q_\xi$; final prediction $\hat{y}_h$.

---

## C    NOTATION TABLE

For clarity, we have attached a notation table here, summarizing all symbols used in the main paper. The paper uses the generic notation $\mathbf{x}$ to denote patient input features. We use plain letters for scalars (e.g., $d, t, y$), boldface lowercase letters for vectors (e.g., $\mathbf{p}, \mathbf{r}, \mathbf{h}$), and boldface uppercase letters for matrices (e.g., $\mathbf{M}$). Euler script letters (e.g., $\mathcal{H}, \mathcal{C}$) denote sets or hierarchies.

Table 5: Notations used in UDONCARE.

| Symbols | Descriptions |
|---|---|
| $x^{(i)}, V_t^{(i)}$ | EHR data of the $i$-th patient; the $t$-th visit of the $i$-th patient |
| $\mathbf{x} = \sum_{t=1}^{T} x_t$ | Aggregated multi-visit representation used for domain discovery |
| $\mathcal{C} = \{c_1, \ldots, c_{|\mathcal{C}|}\}$ | Set of medical concepts (leaf-level ICD-9-CM codes) |
| $y^{(i)} \in \{0,1\}^d$ | Ground-truth label vector for predictive tasks (binary or multi-label) |
| $f_{\phi,k}(\cdot), \mathbf{p}_k$ | Feature encoder for concept type $k$ (diagnosis, treatment, medication) |
| $\mathbf{p}_k = f_{\phi,k}(x_k)$ | The patient embedding for key $k$ |
| $\mathbf{p} = \mathbf{p}_1 \oplus \cdots \oplus \mathbf{p}_K$ | Concatenated patient-level embedding |
| $\mathcal{H}$ | Concept hierarchy with $H$ levels (ICD-9-CM) |
| $n_i^{(h)}$ | Node $i$ at hierarchy level $h$ |
| $\text{Desc}(n)$ | Descendant leaf nodes of node $n$ |
| $e_i, e_i^{(h)} \in \mathbb{R}^h$ | Embedding of concept node or intermediate ontology node |
| $m : c_i \mapsto n_i^{(H)}$ | Mapping from a medical concept to its leaf-level ontology node |
| $S(n)$ | Score for pruning node $n$ based on purity, coverage, and depth |
| $\text{cov}(n), \text{pur}(n), \text{dep}(n)$ | Coverage, purity, and depth used in node scoring |
| $\alpha, \rho$ | Hyperparameters controlling score balance and similarity threshold |
| $\mathbf{M} \in \{0,1\}^{N_{\text{tr}} \times |C'|}$ | Domain-assignment matrix after pruning; encodes soft domain labels |
| $C'$ | Pruned code vocabulary representing discovered domains |
| $g_\theta(\cdot), \mathbf{r} = g_\theta(\mathbf{m})$ | Domain encoder producing latent domain representation $\mathbf{r}$ |
| $\bar{\mathbf{p}} = \mathbb{E}[\mathbf{p}|\mathbf{m}]$ | Mean patient embedding per domain, used for self-supervised alignment |
| $L_r$ | Self-supervised loss (MSE + MMD) for domain encoder training |
| $\mathbf{h} = \mathbf{p} - \tilde{\mathbf{r}}$ | Invariant feature obtained by removing domain-parallel components |
| $\tilde{\mathbf{r}} = \mathbf{r} \cdot \langle \mathbf{p}/\|\mathbf{r}\|, \mathbf{r}/\|\mathbf{r}\| \rangle$ | Parallel projection of $\mathbf{p}$ onto domain direction $\mathbf{r}$ |
| $d_\eta(\cdot), \hat{y}_p$ | Backbone prediction head producing $\hat{y}_p = d_\eta(\mathbf{p})$ |
| $q_\xi(\cdot), \hat{y}_h$ | Invariant prediction head producing $\hat{y}_h = q_\xi(\mathbf{h})$ |
| $\tilde{y}$ | Averaged pseudo-label from $(\hat{y}_p, \hat{y}_h)$ for KL consistency |
| $L_p, L_h$ | Mutual learning objectives combining CE loss and KL divergence |
| $\lambda$ | KL divergence weight. |
| $\langle \cdot, \cdot \rangle$ | Vector inner product |
| $\| \cdot \|$ | Vector norm (Euclidean unless specified) |
| $W_{\text{cls}}(\cdot)$ | the learned weight of a logistic regression task |

## D    BASELINES

Beyond Base and Oracle, we select 5 approaches following general DG setting and select 2 highly related baselines tackling clinical DG problems to compare the performance with UDONCARE:

- Domain-adversarial neural networks (DANN) (Ganin et al., 2016) use gradient reversal layer for domain adaptation, and we adopt it for the generalization setting by letting the discriminator only predict training domains.

- Conditional adversarial net (CondAdv) (Isola et al., 2017) concatenates the label probability and the feature embedding to predict domains in an adversarial way.

- Meta-learning for domain generalization (MLDG) (Li et al., 2018a) adopts the model-agnostic meta learning (MAML) (Finn et al., 2017) framework for domain generalization.

- Invariant risk minimization (IRM) (Arjovsky et al., 2019) learns domain invariant features by regularizing squared gradient norm.

- Proxy-based contrastive learning (PCL) (Yao et al., 2022) build a new supervised contrastive loss from class proxies and negative samples.

- Many-domain generalization for healthcare (ManyDG) (Yang et al., 2023a) with auto-encoder structures to learn invariant features with unique domain separation for each patient.

- Self-learning framework for domain generalization (SLDG) (Wu et al., 2023) discovers latent domains by decoupled domain-specific classifiers for clinical prediction.

Note that, for baselines that rely on domain IDs, we use admission time as the domain definition.

## E    IMPLEMENTATION DETAILS

Considering the common use of the Encoder-Decoder structure for clinical prediction, we adopt the Transformer (Vaswani et al., 2023) as the backbone feature extractor $f_\phi$ in UDONCARE and all baselines. Specifically, we follow the implementation adapted from PyHealth (Yang et al., 2023b), consisting of three layers with a hidden size of 64, 4 attention heads, and a dropout rate of 0.2. The position encoding is applied across patient visits to capture temporal order. Diagnosis, treatment, and medication codes are embedded as separate feature keys using an embedding look-up table. The MLP classifiers used for both original and domain-invariant representations contain two hidden layers with sizes [64, 32], ReLU activations, and a dropout rate of 0.2, with task-specific output activations. We use the Adam optimizer for training, and all remaining hyperparameters follow the settings in PyHealth. For domain discovery, the score function uses $\alpha = 0.5$, and the KL divergence loss coefficient $\lambda$ is set to 1.0 on MIMIC-III and 1.5 on MIMIC-IV. All models are trained for 100 epochs, and the best model is selected based on the AUPRC score monitored on the source validation set. We set the learning rate to $1 \times 10^{-4}$ for $f_\phi$ and $5 \times 10^{-5}$ for $g_\theta$, batch size to 32, iteration to 3, and self-supervised epoch to 30. We tune $\alpha$ and $\lambda$ based on validation performance. All experiments are conducted using Python 3.10 and PyTorch 2.3.1 with CUDA 12.4 on a server equiped with AMD EPYC 9254 24-Core Processors and NVIDIA L40S GPUs.

## F    ADDITIONAL EXPERIMENTAL RESULTS

### F.1    RESULTS WITH GAMENET BACKBONE

We extend the main experiments by replacing the Transformer backbone with GAMENet (Shang et al., 2019b), a model specifically designed for drug recommendation tasks. Therefore, we evaluate its performance on the drug recommendation task using both MIMIC-III and MIMIC-IV. As shown in Table 6, the gap between Oracle and Base again highlights the domain shift between datasets. Standard domain generalization methods (e.g., DANN, IRM) provide limited improvements, while adversarial and meta-learning approaches (CondAdv, MLDG) show moderate gains. ManyDG performs well on MIMIC-IV, but our proposed UDONCARE achieves the best overall results, with consistent improvements in both AUPRC and F1-score. These findings confirm that UDONCARE generalizes effectively even with a domain-specific backbone.

Table 6: **Performance of Drug Recommendation with GAMENet.**

| Model | MIMIC-III | | MIMIC-IV | |
|---|---|---|---|---|
| | **AUPRC** | **F1-score** | **AUPRC** | **F1-score** |
| Oracle | 79.57 (0.32) | 68.14 (0.27) | 75.92 (0.27) | 64.57 (0.36) |
| Base | 72.49 (0.17) | 58.32 (0.21) | 67.88 (0.30) | 58.26 (0.27) |
| DANN | 74.24 (0.09) | 59.21 (0.17) | 72.18 (0.13) | 60.35 (0.18) |
| CondAdv | 73.38 (0.11) | 62.94 (0.13) | 70.86 (0.28) | 59.55 (0.27) |
| MLDG | 75.26 (0.14) | 63.33 (0.16) | 71.49 (0.19) | 61.19 (0.38) |
| IRM | 72.13 (0.21) | 59.87 (0.25) | 71.35 (0.20) | 60.87 (0.34) |
| ManyDG | 76.84 (0.10) | 64.58 (0.30) | 74.18 (0.24) | **62.23 (0.29)** |
| **UdonCare** | **77.56 (0.15)** | **65.79 (0.24)** | **74.83 (0.21)** | 61.94 (0.32) |

## F.2 RESULTS WITH CGL BACKBONE

We also extend the main experiments by replacing the Transformer backbone with CGL (Lu et al., 2021), a model specifically designed for diagnosis prediction tasks. Following the original CGL setting, we only consider conditions as input features, which naturally leads to some performance degradation compared to the main experiment results. Therefore, we evaluate its performance on the diagnosis prediction task using both MIMIC-III and MIMIC-IV. As shown in Table 7, the gap between Oracle and Base again highlights the domain shift between datasets. Standard domain generalization methods (e.g., DANN, IRM) provide limited improvements, while adversarial and meta-learning approaches (CondAdv, MLDG) show moderate gains. ManyDG performs competitively, but our proposed UDONCARE still achieves the best overall results, with consistent improvements across both w-$F_1$ and R@10. These findings confirm that UDONCARE generalizes effectively even with a task-specific backbone.

Table 7: **Performance of Diagnosis Prediction with CGL.**

| Model | MIMIC-III | | MIMIC-IV | |
| --- | --- | --- | --- | --- |
| | w-$F_1$ | R@10 | w-$F_1$ | R@10 |
| Oracle | 25.71 (0.18) | 38.01 (0.21) | 27.09 (0.22) | 39.42 (0.19) |
| Base | 19.05 (0.14) | 30.01 (0.24) | 19.22 (0.25) | 30.64 (0.18) |
| DANN | 20.35 (0.20) | 33.56 (0.27) | 23.02 (0.15) | 34.07 (0.23) |
| CondAdv | 21.55 (0.13) | 35.11 (0.22) | 25.02 (0.28) | 36.51 (0.16) |
| MLDG | 20.12 (0.26) | 32.81 (0.19) | 23.15 (0.19) | 33.27 (0.21) |
| IRM | 21.02 (0.12) | 32.11 (0.25) | 22.41 (0.27) | 33.01 (0.22) |
| ManyDG | 22.01 (0.23) | 35.11 (0.20) | 24.12 (0.17) | 36.02 (0.29) |
| **UdonCare** | **23.89 (0.20)** | **37.12 (0.24)** | **26.02 (0.15)** | **38.21 (0.18)** |

## F.3 RESULTS OF CROSS-INSTITUTIONAL DISTRIBUTIONAL SHIFT

Following the setting used in SLDG (Wu et al., 2023), we extend our evaluation to the eICU dataset, where hospitals can be categorized into four groups based on their locations (Midwest, Northeast, West, and South). To quickly deploy our model on this dataset, we use the processing procedure of PyHealth. We evaluate the performance of our model under spatial domain shifts by treating the Midwest group as the target domain and the remaining groups as the source domain. We use diagnosis and treatment codes to construct patient records, as medications in eICU are not encoded with standard IDs. Table 8 reports the results for readmission and diagnosis prediction. We observe that UDONCARE continues to demonstrate the superiority over all baselines across these two tasks.

Table 8: **Performance under Cross-Institutional Distributional Shift on eICU.**

| Model | Readmission Prediction | | Diagnosis Prediction | |
| --- | --- | --- | --- | --- |
| | AUPRC | AUROC | w-$F_1$ | R@10 |
| Oracle | 22.74 (0.13) | 69.29 (0.13) | 63.18 (0.08) | 79.05 (0.09) |
| Base | 11.97 (0.08) | 51.31 (0.06) | 53.89 (0.05) | 71.40 (0.05) |
| DANN | 12.75 (0.16) | 54.28 (0.14) | 58.41 (0.18) | 75.66 (0.24) |
| CondAdv | 13.16 (0.18) | 53.97 (0.12) | 57.04 (0.23) | 73.92 (0.17) |
| MLDG | 14.82 (0.11) | 54.25 (0.17) | 58.13 (0.28) | 75.31 (0.21) |
| IRM | 15.73 (0.19) | 58.31 (0.21) | 59.22 (0.16) | 76.46 (0.15) |
| PCL | 15.14 (0.23) | 58.06 (0.24) | - | - |
| ManyDG | 17.53 (0.17) | 60.20 (0.10) | 60.61 (0.38) | 76.94 (0.21) |
| SLDG | 16.71 (0.21) | 59.54 (0.18) | - | - |
| **UdonCare** | **18.37 (0.18)** | **62.12 (0.11)** | **61.83 (0.14)** | **77.02 (0.11)** |

Table 9: **Performance comparison of four prediction tasks on MIMIC-III/MIMIC-IV.** We report the average performance (%) and the standard deviation (in bracket) over 5 runs.

| Model | Task 1: Mortality Prediction | | | | Task 2: Readmission Prediction | | | |
| | MIMIC-III | | MIMIC-IV | | MIMIC-III | | MIMIC-IV | |
| | AUPRC | AUROC | AUPRC | AUROC | AUPRC | AUROC | AUPRC | AUROC |
|---|---|---|---|---|---|---|---|---|
| Oracle | 16.54 (0.49) | 70.12 (0.58) | 8.79 (0.50) | 68.61 (0.44) | 73.11 (0.50) | 70.02 (0.47) | 67.51 (0.14) | 67.28 (0.13) |
| Base | 11.62 (0.60) | 55.59 (0.98) | 4.21 (0.45) | 59.02 (0.73) | 50.44 (0.83) | 45.03 (0.68) | 48.18 (0.21) | 45.96 (0.30) |
| UDONCARE | **16.03 (0.35)** | **69.26 (0.40)** | **6.59 (0.29)** | **66.45 (0.46)** | **71.45 (0.33)** | **67.61 (0.36)** | **61.81 (0.11)** | **58.38 (0.23)** |

| Model | Task 3: Drug Recommendation | | | | Task 4: Diagnosis Prediction | | | |
| | MIMIC-III | | MIMIC-IV | | MIMIC-III | | MIMIC-IV | |
| | AUPRC | F1-score | AUPRC | F1-score | w-$F_1$ | R@10 | w-$F_1$ | R@10 |
|---|---|---|---|---|---|---|---|---|
| Oracle | 80.52 (0.13) | 67.48 (0.29) | 74.62 (0.27) | 61.61 (0.23) | 26.53 (0.14) | 39.57 (0.20) | 27.92 (0.12) | 40.68 (0.17) |
| Base | 68.39 (0.14) | 47.82 (0.34) | 66.62 (0.20) | 53.47 (0.20) | 21.69 (0.15) | 31.11 (0.18) | 20.22 (0.13) | 31.65 (0.20) |
| UDONCARE | **78.62 (0.20)** | **66.12 (0.30)** | **72.79 (0.35)** | **59.49 (0.15)** | **24.55 (0.12)** | **38.26 (0.21)** | **27.11 (0.12)** | **39.63 (0.18)** |

## G    DOMAIN DISCOVERY WITH MORE FEATURE KEYS

To examine whether latent domain discovery benefits from richer feature information, we further incorporate procedure and medication codes as additional keys in UDONCARE. As shown in Table 9, the overall gains across prediction tasks are marginal and vary inconsistently across datasets. While certain metrics observe slight improvements, others remain unchanged or even decline. This suggests that introducing conditions, treatments and drugs together into the domain partitioning process does not lead to stable enhancements. These findings are also consistent with our understanding, since we are more likely to get fine-grained domains: patients with the same disease may be split into different domains solely due to distinct medication regimens, which can weaken the benefits of ontology-guided grouping. Therefore, our conclusion is simply that using several ontologies at the same time is less efficient. Instead, we encourage practitioners to select the most appropriate ontology according to the task.

Note that, our reliance on the ICD-9-CM is not due to a limitation of our model, but because diseases are the most direct and required source of patient information. Almost all EHR prediction tasks rely on diagnostic codes as a primary input, and diseases exert strong influence on domain formation. Hence, the disease ontology serves as the most appropriate basis in the general clinical setting. It is also important to note that UdonCare is not restricted to disease ontologies. For instance, using a medication ontology (ATC classification) might be equally or even more appropriate in drug recommendation. Our model directly supports such substitution. In this paper, we want to align the used ontology across four tasks in our experiments.

## H    CONVERGENCE ANALYSIS VIA ITERATIVE LEARNING

While the pruning-based algorithm provides efficiency, its iterative nature makes it non-trivial to characterize convergence using a simple continuous optimization view. To further substantiate the convergence property of our approach, we extend the cosine similarity experiment described in Section 4.3. Specifically, instead of reporting a single snapshot after three iterations, we monitor the cosine similarity values across iterations. As shown in Table 10, the results exhibit a clear trend: at the early stage of training (epoch 40), the similarities between $W_d(\mathbf{p} \rightarrow \text{labels})$ and $W_d(\mathbf{h} \rightarrow \text{labels})$ are relatively low, while the cross-domain similarities ($W_d(\mathbf{p} \rightarrow \text{domains})$ vs. $W_d(\widetilde{\mathbf{r}} \rightarrow \text{domains})$) are comparatively high. This indicates that the model has not yet disentangled domain- and label-related features. As the training proceeds (epoch 60 and 80), the similarities gradually align with the final values at epoch 100, where the decomposition becomes stable and consistent with the results reported in Table 3. These observations provide additional evidence that the iterative learning strategy enables the model to converge in terms of separating label-invariant and domain-specific information. More importantly, by tracking cosine similarity dynamics, we validate that pruning and iterative decomposition jointly lead to a stable representation space, rather than an artifact of a single training snapshot.

Table 10: **Cosine similarity of linear weights on** $\mathrm{p}, \widetilde{\mathrm{r}}$, **and** $\mathrm{h}$ **across epochs.**

| Epoch | Cosine Similarity | Mortality Prediction | Readmission Prediction | Drug Recommendation | Diagnosis Prediction |
|---|---|---|---|---|---|
| 40 | $W_d(\mathbf{p} \to \text{labels})$ vs. $W_d(\mathbf{h} \to \text{labels})$ | 0.5283 | 0.2935 | 0.3829 | 0.6259 |
| | $W_d(\mathbf{p} \to \text{domains})$ vs. $W_d(\widetilde{\mathbf{r}} \to \text{domains})$ | 0.4615 | 0.4908 | 0.5275 | 0.4705 |
| | $W_d(\mathbf{p} \to \text{labels})$ vs. $W_d(\mathbf{p} \to \text{domains})$ | 0.2495 | 0.2695 | 0.2993 | 0.3826 |
| 60 | $W_d(\mathbf{p} \to \text{labels})$ vs. $W_d(\mathbf{h} \to \text{labels})$ | 0.7245 | 0.4119 | 0.6080 | 0.8318 |
| | $W_d(\mathbf{p} \to \text{domains})$ vs. $W_d(\widetilde{\mathbf{r}} \to \text{domains})$ | 0.3960 | 0.2195 | 0.3450 | 0.2712 |
| | $W_d(\mathbf{p} \to \text{labels})$ vs. $W_d(\mathbf{p} \to \text{domains})$ | 0.1842 | 0.1007 | 0.1812 | 0.1437 |
| 80 | $W_d(\mathbf{p} \to \text{labels})$ vs. $W_d(\mathbf{h} \to \text{labels})$ | 0.7761 | 0.4686 | 0.6392 | 0.8523 |
| | $W_d(\mathbf{p} \to \text{domains})$ vs. $W_d(\widetilde{\mathbf{r}} \to \text{domains})$ | 0.3508 | 0.2012 | 0.2807 | 0.2208 |
| | $W_d(\mathbf{p} \to \text{labels})$ vs. $W_d(\mathbf{p} \to \text{domains})$ | 0.1312 | 0.0853 | 0.1296 | 0.1017 |
| 100 | $W_d(\mathbf{p} \to \text{labels})$ vs. $W_d(\mathbf{h} \to \text{labels})$ | 0.7831 | 0.4813 | 0.6546 | 0.8654 |
| | $W_d(\mathbf{p} \to \text{domains})$ vs. $W_d(\widetilde{\mathbf{r}} \to \text{domains})$ | 0.3427 | 0.1957 | 0.2753 | 0.2133 |
| | $W_d(\mathbf{p} \to \text{labels})$ vs. $W_d(\mathbf{p} \to \text{domains})$ | 0.1239 | 0.0794 | 0.1251 | 0.0972 |

$*$ $W_d(\cdot)$ represents the learned linear weights. Cosine similarity scores are averaged across all classes and evaluated over 3 runs.

# I  UPWARD INFORMATION FLOW

To enable hierarchy-aware domain discovery, the model must obtain embeddings for *all* nodes in the ontology $\mathcal{H}$, not only the leaf concepts $\mathcal{C}$. This appendix details the complete procedure for (i) initializing node features, (ii) propagating them upward through the hierarchy, and (iii) refining ancestor embeddings using hierarchical similarity.

**Initializing Leaf-Node Embeddings** First, for each diagnostic code $d_i$, we can initialize its embedding $\mathbf{e}_{d_i} \in \mathbb{R}^h$ in terms of one of two scenarios:

1. For present code $d_i$ in the dataset, $\mathbf{e}_{d_i}$ is initialized by embedding table in disease-specific extractor $f_{\phi,d}(\cdot)$.

2. For absent code $d_i$ in the dataset, $\mathbf{e}_{d_i}$ is initialized by its entity name through the pretrained ClinicalBERT (Huang et al., 2019).

**Bottom-Up Propagation in the Ontology** Since internal nodes have no direct representations, we propagate information from leaf nodes upward. For each parent node $n_i^{(h-1)}$ at level $h-1$, its embedding is initialized by averaging the embeddings of its descendants at level $h$:

$$\mathbf{e}_i^{(h-1)} \;=\; \frac{1}{|\mathrm{Desc}(n_i^{(h-1)})|} \sum_{n \in \mathrm{Desc}(n_i)} \mathbf{e}_n^{(h)},$$

which extends the leaf-level table $\mathrm{E}(e_1, \ldots, e_{|\mathcal{C}|})$ to $\mathrm{E}(e_1, \ldots, e_{|\mathcal{H}|})$. However, these initial embeddings do not capture the hierarchical distances between node pairs. For example, two codes might have significantly different embeddings despite sharing the same parent node.

**LCA-Based Refinement via Hierarchical Similarity** Averaging descendants does not reflect *hierarchical distances*: two nodes may have similar features even if their parents are far apart in $\mathcal{H}$. To encode such structure, we incorporate a hierarchical-clustering refinement (Johnson, 1967).

**(1) Compute pairwise cosine similarity.** For any two nodes $(d_i, d_j)$ (leaf nodes only), we compute

$$\cos(\mathbf{e}_{d_i}, \mathbf{e}_{d_j}) = \frac{\mathbf{e}_{d_i}^{\top} \mathbf{e}_{d_j}}{\|\mathbf{e}_{d_i}\| \, \|\mathbf{e}_{d_j}\|}.$$

Among all possible pairs, we select the one with the highest cosine similarity.

**(2) Update the lowest common ancestor (LCA).** Let $n_{\mathrm{LCA}}(d_i, d_j)$ denote their lowest common ancestor. We update its embedding by averaging itself with the embeddings of the two nodes:

$$\mathbf{e}_{\mathrm{LCA}(d_i, d_j)} \leftarrow \mathrm{Average}\big(\mathbf{e}_{\mathrm{LCA}(d_i, d_j)}, \mathbf{e}_{d_i}, \mathbf{e}_{d_j}\big).$$

**(3) Iterative refinement.** We repeat this process along the ordered similarity list and stop once the maximum remaining similarity falls below a predefined threshold $\rho = 0.3$. Nodes never selected as LCAs retain their initial average-based embeddings from equation 3.

## J    SCORING FUNCTION

The candidate score $S(n)$ involves three indicators:

1. **Coverage**, $\text{cov}(n)$: measures the ratio of covered leaf nodes $n \in \mathcal{L}$. A higher value indicates greater coverage, favoring the selection of higher-level nodes:

$$\text{cov}(n) = \frac{|\mathcal{M}|}{|\mathcal{L}|},$$

   where $\mathcal{M} = \text{Desc}(n)$ denotes the set of descendant leaf nodes of $n$, and $|\mathcal{L}|$ is the total number of leaf nodes in the hierarchy. A higher value indicates that $n$ covers a larger portion of the tree, thus favoring higher-level nodes.

2. **Purity**, $\text{pur}(n)$: gauges how cohesive the partitioned tree is in the embedding space. This metric evaluates how coherent the subtree rooted at $n$ is in the embedding space. It is computed as the expected similarity between the embedding of node $n$ and the embeddings of its descendant leaf nodes:

$$\text{pur}(n) = \mathbb{E}_{m \in \mathcal{M}} \left[\text{sim}(\mathbf{e}_n, \mathbf{e}_m)\right],$$

   where $\text{sim}(\cdot)$ denotes the similarity function (e.g., cosine similarity). Higher purity implies that leaf nodes under $n$ are more semantically aligned with the root node.

3. **Depth**, $\text{dep}(n) \in \{1, \dots, H\}$: indicates the hierarchical level of node $n$. It is defined as:

$$\text{dep}(n) = \frac{h}{H},$$

   where $h$ is the depth of node $n$ and $H$ is the maximum depth of the hierarchy. A larger value corresponds to finer-grained concepts, thus favoring deeper nodes.

## K    BROADER APPLICABILITY AND EXTENSIONS OF UDONCARE

While UDONCARE is evaluated primarily on structured, code-based EHR data, the core design of the framework is modality-agnostic and can be extended beyond the four clinical event prediction tasks explored in this work. We briefly highlight two promising directions:

### K.1    EXTENSION TO MULTI-MODAL CLINICAL DATA

The ontology-guided domain discovery mechanism is compatible with richer, multi-modal patient records. Modern hospital systems routinely collect diverse modalities, including:

1. Clinical notes (text), encoded with contextualized language models;

2. Radiology and pathology images, structured under well-defined ontologies such as RadLex;

3. Physiological waveforms (ECG, EEG), which contain temporally structured signals;

4. Laboratory trajectories, which follow standardized concept hierarchies (LOINC).

In principle, our model can incorporate these modalities by (1) extending the feature extractor to relevant encoders (e.g., CNNs, transformers for images, waveform encoders), and (2) pruning cross-modal ontologies or multi-level taxonomies whenever available. The mutual-learning mechanism in UDONCARE, designed to decouple domain and label information, benefits from such multi-modal richness by preventing modality-specific biases from dominating the latent domain assignments. However, large-scale multi-modal extensions require access to paired datasets, which is unavailable in MIMIC or other standard public EHR databases. Thus, multi-modal generalization remains an important but data-dependent direction for future research.

### K.2    GENERALIZATION BEYOND HEALTHCARE TO ONTOLOGY-RICH DOMAINS

The ontology-guided design is also not restricted to clinical settings. Many real-world domains exhibit hierarchical or taxonomy-driven feature structures that mirror the ICD hierarchy, such as

1. E-commerce: product category graphs, brand hierarchies, user segments;

2. Recommendation systems: taxonomies of items or content genres;

3. Knowledge-driven applications such as academic topic hierarchies, legal codes, or biological ontologies.

In these domains, domain shifts commonly arise from evolving user groups, temporal drifts in product popularity or region-specific differences, which are analogous to the clinical temporal and spatial shifts studied in this work. The pruning-based latent domain discovery can meaningfully identify stable high-level clusters across such hierarchies, while the invariant-projection mechanism can separate domain-specific variations from task-relevant signals. The adaptability of its ontology-guided components suggests broad potential applicability to many hierarchical, knowledge-grounded machine learning scenarios.

## L    MORE EXPLANATIONS ON THE LEARNED LINEAR WEIGHTS

In this section, we discuss the relatively low cosine similarity in Table 3:

First, a similar trend has been observed in ManyDG (Yang et al., 2023a). Linear probing for domain prediction is inherently more challenging than probing for label prediction. The domain classification task usually contains a substantially larger number of categories (e.g. over 500), whereas the label space is considerably smaller. Thus, the learned linear coefficients for domain prediction tend to be less aligned, resulting in lower cosine similarity in the second row.

Second, if the model successfully decomposes patient embeddings into label-related and domain-related components, the observed similarity pattern becomes intuitive. The first row corresponds to label prediction, and thus naturally shows higher similarity because label information constitutes a larger and stronger portion of the embedding space. In contrast, after label (domain-invariant) information is accounted for, the remaining domain signal becomes weaker, making it harder for a linear classifier to capture consistently. This yields lower similarity in the second row.

We emphasize that this explanation does not imply a precise quantitative ratio between label and domain information. Rather, the key observation is that when the invariant (label-relevant) component dominates the embedding space, the remaining domain-specific signal becomes harder to recover linearly, and this phenomenon is reflected consistently in our empirical results.

## M    THE USE OF LARGE LANGUAGE MODELS (LLMS)

In preparing this paper, we used large language models (LLMs) solely as a general-purpose tool to improve writing fluency and polish the presentation of the text. All ideas, experimental designs, analyses, and conclusions are our own, and the responsibility for the content rests entirely with the authors.

