# OpenReview forum: "Hierarchy Pruning for Unseen Domain Discovery in Predictive Healthcare"
_ICLR.cc/2026/Conference — Submitted to ICLR 2026_

### Official Review · Reviewer_J8r2 · 2025-10-27

**Soundness:** 1
**Presentation:** 2
**Contribution:** 2
**Rating:** 2
**Confidence:** 3

**Summary:**

UdonCare attempts to improve accuracy with various medical predictions by using domain generalization techniques to better predict across a wide range of conditions.

**Strengths:**

- Evaluations on MIMIC-III and MIMIC-IV are good
- AUROC and AUPRC are really good metrics for model evaluation
- Task selection is also good. Mortality, Readmission, Drug Recommendation, and Diagnosis Prediction are all important.

**Weaknesses:**

- The list of baselines is very very incomplete. In particular, this paper is critically missing the most important structured EHR data baseline: gradient boosted trees (either through xgboost or lightgbm). I would recommend reject purely for that missing baseline alone as it is such an important baseline in EHR settings.
- Time splitting is a good approach, but the way it is done in this paper is incorrect. This paper uses as the test set all patients with a last visit after 2017 and excludes them from the training set. This is wrong. The visits for these patients which were before 2017 should be in the training set. The reason for this is because the train test split needs to approximate a realistic train/test deployment in the model. We right now don't know which patients are going to get visits in the future so we can't exclude them from the training set for a deployment right now. So you shouldn't exclude patients with future visits from a model backtesting procedure.
- The base models getting worse than random accuracy on AUROC for readmission prediction is very very suspicious, and I think an indication that something went wrong with your setup. This might be because I didn't understand your target data vs source data distinction. What is your target data and what is your source data? Is it the time split.
- The source and target domains for table 2, your main experiments. I really confusing. What they precisely?

**Questions:**

I just want to confirm: the "target data" in your experiments section is the 2017 data? And the "source data" is the earlier data? Or am I misunderstanding something?

And this target data is unused for all models (except oracle), including your domain generalization models?

---

> ### Author Response · Authors · 2025-11-21
>
> We sincerely appreciate the reviewer's valuable feedback, which has provided us with excellent opportunities to improve our work. In the following section, we address each comment in detail.
>
> **W1: Lack comparisons with baselines like XGBoost and LightGCM**
>
> We acknowledge the historical role of gradient-boosted trees in EHR prediction. However, we would like to clarify why we didn't use them in baselines:
>
> - From the perspective of clinical prediction, most clinical prediction studies (especially after 2020) only rely on and compare with deep sequential baselines such as **RNN/CNN-based**, **Transformer-based**, and **Graph(typically GNN)-based** models. Note that, such categorization is not self-defined and is used by most previous works focusing on addressing EHR prediction problems [1,2,3,4]. These models are used because they can (i) learn medical-concept embeddings, (ii) model temporal visit sequences, and (iii) incorporate medical knowledge structures. It is also the motivation why we select Transformer, GAMENet (drug graph), CGL (disease graph) as backbone models in experiments. In contrast, gradient-boosted trees cannot effectively capture longitudinal visit dependencies or learn patient representations from code embeddings, and thus they are no longer treated as competitive baselines in most papers.
> - From the domain generalization (DG) perspective, boosted trees are incompatible with DG baselines, which either modify or regularize learned hidden representations. In contrast, gradient-boosted trees can be considered as a classifier and cannot participate in DG training objectives. To the best of our knowledge, XGBoost and LightGCM are not typical choices as either backbone or baselines for DG problems in literature. For instance, in image scenes, most works focus on CNN/ViT as backbone, and compare their models with other DG baselines (e.g. MLDG).
>
> **W2: For time splitting, the visits for patients (last visit after 2017) which were before 2017 should be in the training set.**
>
> We agree that your suggestion is indeed one valid way to reflect a realistic deployment scenario. However, in the context of EHR modeling, it might introduce a severer risk of **information leakage**. If a patient later becomes part of the target set, then their historic visits would appear in the training set. Most models can leverage this partial patient trajectory to make target predictions easier, thereby inflating predictive performances.
>
> Hence, we use the conservative evaluation that better reflects the model generalization. We also note that similar temporal splits have been used in SLDG [5] (Section 4.1 Data Split): "Patients admitted after 2014 are used as the target testing data."
>
> **W3: The worse AUROC of Base model for readmission prediction**
>
> We clarify that it mainly stems from the source–target distinction.
>
> As explained in our response to W2, we remove records of patients whose last visit occurs after 2017 from source subset, so both source (training) and target (test) set contain disjoint patient populations. This creates a much larger distribution shift than in standard data splits. Under this conservative setup, the Base (e.g. Transformer only) model tends to overfit the source domain. When evaluated on a target distribution it has never encountered, this model will have low AUROC < 0.5 and model generalization without DG consideration.
>
> We will strengthen the explanation in the paper to avoid confusion. Your comment helped us better articulate the implications of this strict evaluation setup! :)
>
> **W4 & Q1: What the source and target domains in main experiment (Table 2) are? Is target data visible for DG baselines (except Oracle)?**
>
> Most of your current understanding aligns with our setup. The source and target domain used in main experiments correspond exactly to the train/test split described in the Data Split section. The results reported in Table 2 are the performance evaluated only on the target domain (test set).
>
> The target data is not visible to DG baselines (except oracle) during training. In simple terms, the only difference is that DG methods learn domain-invariant features (which can be consider as a regularization), so that the model does not overfit to the training data. As a result, DG can have better model generalization, especially when the distribution shift is large between training and test sets.
>
> ---
>
> *Reference:*
>
> *[1] Collaborative graph learning with auxiliary text for temporal event prediction in healthcare, IJCAI 2021*
>
> *[2] Graphcare: Enhancing healthcare predictions with personalized knowledge graphs, ICLR 2024*
>
> *[3] Graph transformers on EHRs: Better representation improves downstream performance, ICLR 2024*
>
> *[4] Self-supervised graph learning with hyperbolic embedding for temporal health event prediction, IEEE Cybernetics 2021*
>
> *[5] An iterative self-learning framework for medical domain generalization, NeurIPS 2023*

---

> ### Comment · Reviewer_J8r2 · 2025-11-22
>
> Thank you for the comprehensive response.
>
> 1. I disagree very strongly with the claim that gradient boosted trees are not relevant baselines anymore. https://arxiv.org/html/2508.12104v1 is a good recent example paper showing very strong performance for gradient boosted trees relative to other models, with the gradient boosted tree exceeding the performance of their sophisticated transformer model in many tasks.
>
> Even though gradient boosted trees have limitations for understanding sequences, they are highly effective on structured data, which in some ways counterbalances their internal limitations.
>
> They also have a very important role as "an easy baseline" to run, which requires relatively minimal tuning or time compared to more sophisticated methods.
>
> 2. If you want to ensure that no patients are shared between train and test, you can enforce that separately with random sampling. Regardless, your sampling strategy is biased.
>
> (Note that I would argue that patient overlap between train and test is ok as it is representative of many model deployment scenarios where models are trained/tuned in the hospital before deployment. I understand that might be controversial, and I would not fault someone for disagreeing. However, I think it is clear that the way that time splitting was done in this paper introduces bias).
>
> 3. I find this surprising, although I do agree it's plausible. I would trust it much more if we had a simple baseline like xgboost/lightgbm that requires less tuning/setup.
>
> 4. Thank you for the clarification, but I would recommend updating the text of the paper to make this more clear. However, now I'm even more confused about the oracle model. Why doesn't the oracle model converge to 100% accuracy (overfitting) if it has access to both train and test?

---

> > ### Author Response · Authors · 2025-11-26
> >
> > Thanks for the reviewer's prompt response.
> >
> > To satisfy the reviewer’s curiosity, we trained XGBoost on the MIMIC-III readmission task and obtained 34.97% AUPRC and 51.84% AUROC. We observe that AUPRC is significantly worse than Base Transformer (50.13%). More importantly, **XGBoost cannot serve as a backbone in the DG settings** (as explained in our previous response), and thus cannot be involved in our experiments.
> >
> > ---
> >
> > **R1:** We carefully read the entire paper the reviewer referenced, including the majority of its experimental results, and we must respectfully but firmly clarify that this preprint paper does not support the reviewer’s claim:
> >
> > 1. The referenced paper (posted **Aug 16, 2025**; last revised Nov 7, 2025) has **not been accepted by any peer-reviewed venue**, and per the ICLR 2026 guidelines, non-peer-reviewed, very recent (<=4 months) works should not be considered as grounds for mandatory baselines. Therefore, using this manuscript to demand additional baselines is not appropriate.
> > 2. This paper does **not** propose or emphasize gradient-boosted trees; its main contribution is a family of **Transformer models** (62M, 119M, 1B parameters). Thus, the paper does not support the reviewer’s claim about the relevance or strength of tree-based models.
> > 3. The only table, Table 15 (p.41), directly comparing the gradient boost tree also shows the Transformer outperforming XGBoost [**MAE: 3.34** (Transformer) **< 3.57** (XGBoost) < 3.67 (Linear Regression)], which contradicts the reviewer’s assertion that gradient-boosted trees “show very strong performance relative to other models.”
> >
> > In summary, the referenced paper does not support the reviewer’s argument, and it even provides evidence that Transformer outperforms over XGBoost in EHR (Length-of-Stay) prediction. We are confused by this irrelevant reference and respectfully ask the reviewer to adhere to the ICLR 2026 Code of Conduct and avoid technology-biased comments.
> >
> > **R2:** The reviewer’s argument contains a clear **naturalistic fallacy**: from the premise that “patient overlap may occur in certain deployment scenarios,” it does **not** logically follow that “our time-splitting introduces bias.” The first statement only implies that overlap can be acceptable, but it cannot be used to conclude that non-overlap causes bias. These are not logically connected claims.
> >
> > In our previous response, we never argued that patient overlap is wrong; we simply stated that **avoiding overlap is preferable** in our context, and we provided two explicit reasons: (1) it mitigates information leakage risk in longitudinal EHR data; and (2) it is also consistent with prior clinical DG literature. The reviewer’s reply does not address these points and instead restates a personal preference without engaging with our rationale.
> >
> > We would appreciate it if the reviewer could engage with our responses using **evidence-based reasoning**. Additionally, could the reviewer please elaborate on how patient-level non-overlap can be enforced under random sampling?
> >
> > **R3:** Thanks for the reviewer's agreement. However, as we explained at the previous response, DG is fundamentally a deep-learning–oriented problem setting, and gradient-boosted trees cannot serve as a backbone in DG. Thus, we are genuinely puzzled by the reviewer’s request in DG settings where such models are not applicable.
> >
> > **R4:** We think that this question is out of the scope of this paper, but we are happy to clarify. When the dataset is not small, individual patients often exhibit heterogeneous patterns that a model cannot perfectly perform. Even if a model is trained and tested on the same temporal-prediction dataset, no ML/DL model can be guaranteed to reach 100% accuracy. Therefore, it is entirely expected that an oracle model in a realistic EHR prediction setting does not converge to 100%.
> >
> > ---
> >
> > By following the reviewer's suggestion in R4, we added two additional sentences to the Data Split and Baseline sections, respectively:
> >
> > > Note that, to avoid information leakage in clinical modeling, we ensure that no patient’s earlier visits appear in the training set if their later visits are included in the test set.
> >
> > > Note that, all baselines follow the same source/target definition as in the Data Split section, and the test set remain unseen to all models except Oracle. The results reported in following sections are the performance evaluated on the test set.

---

> ### Comment · Reviewer_J8r2 · 2025-11-26
>
> Thank you for the comprehensive response, and for running that xgboost baseline.
>
> I'll respond to all of your feedback as well, but for the sake of time I just want to start by personally reproducing your xgboost number, since that seems like a weird result to me. (I already have MIMIC-III downloaded on my machine, so this should in theory be a very straightforward process).
>
> If I can reproduce that terrible xgboost performance, I will probably change my vote to accept as that would resolve most of my concerns.
>
> I have downloaded your code and started by running your run_preprocess.py script.
>
> However, I'm very confused about the train/test split. It seems like your run_preprocess.py script does a random split, not the time split you mentioned? How do I reproduce your time split?

---

> ### Comment · Reviewer_J8r2 · 2025-11-26
> **Critical bug in data processing code**
>
> Dear authors,
>
> While waiting for your response, I decided to take a second look at the data processing code and I think I have identified the bug that is causing all of the problems.
>
> **I have downgraded my score to 0 since this bug appears to be critical**
>
> The problem is the following line in run_preprocess.py:
>
> > patient_admission, admissions = parser.parse(sample_num)
>
> This line of code should have been:
>
> > patient_admission, admissions = parser.parse(sample_num, sorting=True)
>
> This is important because your admission processing requires sorting, but it defaults to false.
>
> This breaks your readmission label entirely, resulting in random noise.
>
> Output from posted (incorrect code)
> > Calculating readmission statistics...
> > Total number of admission intervals: 10828
> > Number of readmissions (<15 days): 5718
> > Readmission rate: 52.81%
>
> Note how high this supposed readmission rate is. Over 50% is very extreme for 14 day readmission!!!
>
> When this bug is fixed, this changes to:
> > Calculating readmission statistics...
> > Total number of admission intervals: 10828
> > Number of readmissions (<15 days): 828
> > Readmission rate: 7.65%
>
> This looks more reasonable.
>
> Authors, can you help explain what is going on? Did you upload the wrong code?

---

> > ### Author Response · Authors · 2025-11-26
> >
> > Thank you for scrutinizing our code and pointing out this issue. **To clarify, all reported results were obtained with sorting=True**. The code included in the paper is intended solely as a concise demonstration, and the exposed option is provided for flexibility; it does not reflect the exact configuration used in our experiments. Therefore, this concern does not affect the validity of our reported results in any way.
> >
> > We also want to emphasize that the provided code was prepared within a very short timeframe in direct response to your question. We apologize for inadvertently uploading a version that contained settings used during local testing. However, **it is not justified to reduce the evaluation score to zero based on such a minor and non-impactful detail**—particularly when it has no bearing on the correctness or significance of our technical contributions.
> >
> > To further assist your verification, we have also included the code used to implement XGBoost (add `xgboost.ipynb` in supplementary materials, which is directly runnable without change on other files), which clearly shows that we sorted the admissions. You may also set sorting=False and observe the outcome; in our tests, the unsorted setting yields results around 70% AUPRC and 65% AUROC, confirming that the reported performance cannot be reproduced without sorting.
> >
> > > When sorting=True, Evaluation: AUPRC: 0.2356 --- AUROC: 0.5769 --- F1: 0.0933 --- Kappa: 0.0485;
> > >
> > > When sorting=False, Evaluation: AUPRC: 0.7139 --- AUROC: 0.6598 --- F1: 0.6818 --- Kappa: 0.2132;

---

> ### Comment · Reviewer_J8r2 · 2025-11-26
>
> I agree that your score should be based on the version of the code that was used for your experiments. However, that code does not appear to be uploaded.
>
> As of the time of writing this reply, run_preprocess.py still has the bug in it.
>
> Even your xgboost jupyter notebook that you just uploaded doesn't include your time splits.
>
> I can only review what is available to me.
>
> Can you please upload a correct version of your code that was used to generate your results?

---

> ### Comment · Reviewer_J8r2 · 2025-11-26
>
> Oh, one final thing. I want to clarify that the priority here is the ability to reproduce at least some of the numbers in table 2 of the original paper. In particular, the two critical ones for reproducing are your base model and udoncare.
>
> Xgboost numbers are both less important and I understand you might not have had time to write code to correctly train an xgboost model.

---

> ### Comment · Reviewer_J8r2 · 2025-11-26
>
> Apologies for the repeated comments, but this should be the last one.
>
> I just realized you forgot to upload your training and evaluation scripts as well. I would appreciate it if you could also ensure the hyperparameters and everything else are set correctly in those scripts so that your results can be easily reproduced.

---

> ### Author Response · Authors · 2025-11-28
>
> We appreciate the reviewer’s engagement throughout the discussion. After carefully reviewing the reviewer’s comments and the subsequent exchanges, we would like to highlight several inconsistencies and unresolved issues that have made it challenging to understand the reviewer’s evaluation criteria:
>
> - **Inconsistency across statements:** The reviewer initially stated that **the absence of gradient boosted models (GBM) alone would justify rejection** [1, 2, 3, 4]. However, after we provide valid xgboost results, the reviewer later asserted that **xgboost is less important** [5].
>
> - **Inconsistency in Implementation:** The reviewer first emphasized the **need to reproduce xgboost** [6]. Later, after we provided the requested baseline and clarified the sorting issue [11], the reviewer stated that **xgboost was no longer the priority** and that instead our Table 2 should be used for reproducibility [5, 7].
> - **Use a Recent, Unrelated, Unpublished Paper:** The reviewer cited a preprint last revised on Nov. 7, 2025 as key evidence for the importance of GBM with confident statement [8]. However, this unpublished paper is **neither related to xgboost, nor showing strong results of xgboost**. This appears inconsistent with the reviewer’s claims, and despite our request for clarification, no response was provided.
>
> These points indicate that, after we addressed each concern, the reviewer did not base their discussion on those clarifications, but instead introduced new minor issues (e.g. data-processing details or run GBM). We have addressed this issue by uploading the XGBoost code and specific configurations. Any further iterations on low-level implementation questions would likely consume additional time without affecting the reviewer’s assessment. More importantly, we are concerned about the reviewer's **self-contradictory statements and claims with insufficient evidence** as shown above.
>
> ICLR 2026 does not require fully reproducible end-to-end code at under-review time. Our supplementary material clearly states that we provide (i) the development model definition (the /models/ directory referenced in the README), and (ii) a draft preprocessing script (not mentioned in the README), with full tested code planned for release upon acceptance. We believe our results can be reproduced by the given /models/ folder.
>
> Until now, the reviewer **still didn't directly reply to our rebuttal** (e.g. whether any of their concerns is solved, why such an irrelevant arxiv paper is referenced with strong statement [8], and why they think our temporal split introduce bias [9]). Furthermore, the reviewer **never** explain **why they ask irrelevant questions out of the paper scope** [12] and **why they provide some personal claims** without evidence or logistic support [1, 2, 3, 6, 8, 9]. We remain unsure how these affect the scientific evaluation of our work.
>
> In summary, the core contributions of our paper have been acknowledged by the other reviewers (e.g. "The paper’s core strength is its novel use of medical ontologies to solve a DG problem"), and no one else has raised concerns about GBM. Since **the most critical issue leading the reviewer's low score has also been resolved** [10, 11], we hope this clarification helps.
>
> ---
>
> ***References***:
>
> *[1] "The list of baselines is very very incomplete."*
>
> *[2] "this paper is critically missing the most important structured EHR data baseline: gradient boosted trees"*
>
> *[3] "I would recommend reject purely for that missing baseline alone"*
>
> *[4] "I disagree very strongly with the claim that gradient boosted trees..."*
>
> *[5] "Xgboost numbers are both less important and I understand you might not have had time to write code to correctly train an xgboost model."*
>
> *[6] "I would trust it much more if we had a simple baseline like xgboost ..."; "If I can reproduce that terrible xgboost performance ..."*
>
> *[7] "I want to clarify that the priority here is the ability to reproduce at least some of the numbers in table 2 of the original paper."*
>
> *[8] " {external_url} is a good recent example paper showing very strong performance for gradient boosted trees relative to other models, with the gradient boosted tree exceeding the performance of their sophisticated transformer model in many tasks."*
>
> *[9] "Note that I would argue that patient overlap between train and test is ok as it is representative of many model deployment scenarios where models are trained/tuned in the hospital before deployment. ...,  However, I think it is clear that the way that time splitting was done in this paper introduces bias."*
>
> *[10] "I have downgraded my score to 0 since this bug appears to be critical"*
>
> *[11] "I agree that your score should be based on the version of the code that was used for your experiments."*
>
> *[12] "However, now I'm even more confused about the oracle model. Why doesn't the oracle model converge to 100% accuracy (overfitting) if it has access to both train and test?"*

---

### Official Review · Reviewer_o6w5 · 2025-10-28

**Soundness:** 3
**Presentation:** 2
**Contribution:** 2
**Rating:** 4
**Confidence:** 4

**Summary:**

This paper investigates how hierarchical medical knowledge can be effectively incorporated into the problem of domain generalization. To this end, the authors introduce UdonCare, a hierarchy-guided framework that iteratively partitions patients into latent domains for domain discovery and disentangles domain-invariant representations from patient data. The proposed framework is empirically evaluated on the MIMIC-III and MIMIC-IV datasets, demonstrating both its effectiveness and computational efficiency.

**Strengths:**

**S1.** The paper tackles a practically important and clinically meaningful problem by investigating how hierarchical medical ontologies can be incorporated into clinical domain generalization (DG) tasks.

**S2.** The proposed UdonCare framework integrates medical ontologies into an iterative domain discovery process. Its design involves a hierarchy-guided domain discovery algorithm, along with distinct domain and backbone pathways that jointly contribute to disentangling domain-invariant representations from patient data.

**S3.** The experimental evaluation includes comparisons with established baseline models, analyses of the decomposition effectiveness, ablation studies, examinations of training data size effects, and runtime analyses, thereby providing a reasonably thorough empirical validation of the proposed framework.

**Weaknesses:**

**W1.** The paper motivates its contribution by highlighting the limited attempts to integrate medical knowledge into DG settings, despite existing works on knowledge-driven predictive modeling and clinical DG methods. While this motivation is reasonable, the paper would benefit from a clearer articulation of the technical challenges involved in bridging these two research lines. Specifically, it should elaborate on what makes the incorporation of hierarchical medical knowledge into DG settings non-trivial and how UdonCare effectively addresses these challenges. Without such clarification, the methodological novelty and technical depth of the proposal appear somewhat limited, particularly since UdonCare relies heavily on established techniques in its framework design.

**W2.** The current scope of UdonCare is limited to modeling relational clinical data, such as diseases, procedures, and medications, within binary classification settings. It would strengthen the paper to discuss the framework’s potential generalization to other data modalities and to broader predictive tasks beyond binary classification.

**W3.** The evaluation of UdonCare is conducted solely on the MIMIC-III and MIMIC-IV datasets, which are relatively homogeneous even after removing overlapping time ranges during preprocessing. To more rigorously assess the generalizability of the proposed framework, it would be beneficial to include experiments on additional datasets with greater heterogeneity, such as those adopted in ManyDG or SLDG.

**W4.** Since the primary objective of UdonCare is domain discovery aimed at enhancing the performance of domain generalization tasks, it would be valuable to provide a deeper analysis of the discovered domains. In particular, interpreting these latent domains through the lens of medical ontology knowledge could yield meaningful insights into their clinical relevance and the underlying patterns captured by the framework. Such analysis would not only substantiate the effectiveness of the proposed discovery process but also strengthen the interpretability and practical utility of the proposal.

**W5.** The overall presentation of the paper could be improved in several aspects to enhance readability and coherence:

* **Abstract:** The connection between cohort modeling and DG should be articulated more explicitly to clarify the paper’s conceptual motivation.

* **Organization:** It would aid readers’ comprehension if the related work section, particularly the discussion of existing DG methods, could be placed earlier in the paper.

* **Preliminaries:** A concise yet precise formulation of the DG problem should be introduced in the preliminaries to provide a clear foundation for the subsequent methodology.

* **Methodology:** The methodological exposition could be streamlined by defining the core mechanisms of UdonCare as distinct modules and elaborating on them in a structured, module-wise manner.

* **Notation:** Given the large number of symbols and variables used, including a notation table summarizing the key notations would substantially improve clarity.

**Questions:**

Beyond W1-W5, I have the following questions for clarification:

**Q1.** The underlying rationale behind the node scoring formulation in Equation (5) should be explained in greater detail.

**Q2.** In Section 4.1, the criteria “$d < 120$” and “$d > 4500$” are mentioned. Could the authors clarify whether this variable $d$ corresponds to the same linear classifier described in Section 4.3? If not, the distinction between the two should be made explicit to avoid confusion.

**Q3.** In the experiment assessing the effectiveness of decomposition (Table 3), it is stated that the first two rows exhibit higher similarities. However, the second row seems to display a notably lower similarity than the first. Additional clarification is needed on this phenomenon and how these results support the conclusions drawn from this experiment.

---

> ### Author Response · Authors · 2025-11-21
> **Official Comment by Authors (Part 1/2)**
>
> We appreciate your detailed and constructive feedback.
>
> **W1: The paper would benefit from articulating what makes the incorporation of hierarchical knowledge into DG non-trivial and how UdonCare addresses these challenges.**
>
> Below we summarize why this problem is non-trivial and how UdonCare addresses these challenges:
>
> - **From a clinical perspective**, existing latent-domain DG models can partition patients in feature space but lack clinical interpretability. In clinical settings, patient cohorts should be defined according to clinically meaningful rules, not just feature similarity. This motivates the use of medical ontologies, which encode long-validated hierarchical structures. Leveraging ontologies as constraints for domain discovery is a direction that has not been explored in prior DG research.
>
> - **From a DG methodology perspective**, incorporating ontologies is non-trivial for two key reasons:  (1) Ontologies cannot be directly adapted into general DG models, which typically rely on clustering in patient embedding space, and the multilevel ontologies cannot make naive integration; (2) In clinical prediction, prior works primarily use ontologies to enrich code embeddings via GNNs or other parameterized encoders, but this does not solve the problem of defining patient cohorts for DG. For separating patient domains, we need it to serve as a structural rule that constrains how patient groups are formed.
>
> UdonCare addresses these challenges by introducing a rule-driven pruning algorithm that adapts the hierarchical ontology into a coherent set of latent domains. Combined with iterative training and rectified beam search, this enables efficient and clinically grounded domain discovery.
>
> We will incorporate these technical challenges in the introduction (and/or in appendix).
>
> **W2: Discuss the framework’s potential generalization to other data modalities and to broader predictive tasks**
>
> First, most EHR data, especially public datasets, are stored in the form of medical codes. As a result, the majority of clinical prediction tasks are built upon code-based inputs. To assess generalization across different predictive settings, we designed our experiments around four vital tasks: readmission & mortality (binary classification), and drug recommendation & diagnosis prediction (multi-label classification). These tasks cover the major problem types in code-based EHR modeling and demonstrate that UdonCare operates effectively under most scenarios.
>
> Second, we agree that multi-modal prediction is an promising direction. From our perspective, UdonCare can be extended to multimodal clinical prediction by incorporating clinical notes (text), radiology images, ECG waveforms, and other modalities. However, such extensions require broader data-access permissions, and most public EHR datasets cannot provide sufficient paired patient records for training. We consider this an exciting avenue for future work.
>
> We also note, as Reviewer 7SsM pointed out in Question 1, that the ontology-guided design of UdonCare is not limited to clinical data. The framework can be applied to other ontology-rich domains such as e-commerce or other knowledge-guided applications, highlighting its general potential beyond healthcare. We will include this discussion in the appendix as part of our further discussion section.
>
> **W3:  Include experiments on additional datasets with greater heterogeneity**
>
> Following the setting used in SLDG [1], we extend our evaluation to the eICU dataset, where hospitals can be categorized into four groups based on their locations (Midwest, Northeast, West, and South). To quickly deploy our model on this dataset, we directly use the processing procedure of PyHealth [3]. We evaluate the performance of our model under spatial domain shifts by treating the Midwest group as the target domain and the remaining groups as the source domain. We use diagnosis and treatment codes to construct patient records, as medications in eICU are not encoded with standard IDs.
>
> The table below reports the results for readmission and diagnosis prediction. We observe that our model continues to demonstrate superiority over the baselines. We will include this additional experiment in the appendix of the revised version.
>
> |Models|AUPRC (Readmission)|AUROC (Readmission)|w-F1 (Diagnosis)|R@10 (Diagnosis)|
> |--------|-----------|-----------|-----------|-----------|
> |Oracle|22.74 (0.13)|69.29 (0.13)|63.18 (0.08)|79.05 (0.09)|
> |Base|11.97 (0.08)|51.31 (0.06)|53.89 (0.05)|71.40 (0.05)|
> |DANN|12.75 (0.16)|54.28 (0.14)|58.41 (0.18)|75.66 (0.24)|
> |CondAdv|13.16 (0.18)|53.97 (0.12)|57.04 (0.23)|73.92 (0.17)|
> |MLDG|14.82 (0.11)|54.25 (0.17)|58.13 (0.28)|75.31 (0.21)|
> |IRM|15.73 (0.19)|58.31 (0.21)|59.22 (0.16)|76.46 (0.15)|
> |PCL|15.14 (0.23)|58.06 (0.24)|-|-|
> |ManyDG|17.53 (0.17)|60.20 (0.10)|60.61 (0.38)|76.94 (0.21)|
> |SLDG|16.71 (0.21)|59.54 (0.18)|-|-|
> |**UdonCare**|**18.37 (0.18)**|**62.12 (0.11)**|**61.83 (0.14)**|**77.02 (0.11)**|

---

> ### Author Response · Authors · 2025-11-21
> **Official Comment by Authors (Part 2/2)**
>
> **W4: Provide a deeper analysis of the discovered domains.**
>
> We agree that showing discovered domains would enhance both interpretability and practical value of our model. We will include one more case study to focus on the pruned disease nodes and analyze the resulting latent domains. This qualitative analysis will help clarify the clinical-relevant patterns captured by the discovery process. We appreciate your suggestion, and we believe this addition will strengthen the paper.
>
> **W5: Suggestions for paper presentation**
>
> - **Abstract:** We will add one sentence bridging cohort modeling and DG before introducing our model;
> - **Related Work & Preliminaries:** We agree that introducing a concise formulation of DG problems in the preliminaries would improve readability. To maintain coherence among first three sections, we prefer to keep the Related Work section near the end of the paper. We will nevertheless revise the transitions to ensure smoother comprehension for readers.
> - **Methodology:** As also noted by Reviewer eoYo in W1–4, we will refine the methodological exposition by providing clearer modular structure and additional details in both the main text and the appendix in the revised version.
> - **Notation:** We will include a notation summary table in the appendix.
>
> **Q1: Equation (5) should be explained in greater detail**
>
> We agree that Equation (5) would benefit from a more detailed explanation, particularly scoring components (coverage, purity, and depth). Due to space limitations in the main paper, we condensed this part of the description.
>
> We will add more details for the node scoring subsection in appendix, and will let you know when revised version uploaded.
>
> **Q2: whether variable $d$ in Section 4.1 corresponds to the same linear classifier described in Section 4.3**
>
> Although both use the same output dimension, they represent different entities (one is a dimensionality parameter, and the other is a decoder classifier). To avoid this ambiguity and following your recommendation, we will rename $W_d(\cdot)$ to $W_{\text{cls}}(\cdot)$. Thank you for catching this, and we will update the notation in the revised version!
>
> **Q3: Clarify why the second row in Table 3 seems to display a notably lower similarity than the first.**
>
> We agree that the similarity pattern in Table 3 needs further clarification, which can be explained from two perspectives:
>
> - A similar trend has been reported in Appendix B.8 of ManyDG [2], which also studies clinical DG problems. Linear probing for domain prediction is naturally more challenging than label prediction, because the domain identification task typically involves much more categories (over 500) compared to the label space. As a result, the learned coefficients in the second row are less similar than those learned in the first row.
> - If we assume that our model successfully separates patient embeddings into a label and a domain component, this pattern becomes interpretable. The first row (label) shows higher similarity, since the label information forms a more substantial portion in patient embeddings. In contrast, a linear classifier would have difficulty capturing the remaining weaker signal, producing lower similarity in the second row. We emphasize that this explanation does not assert the exact proportion of label or domain information, but the observed trend is consistent when invariant part dominates the embedding space.
>
> We will add these additional explanations into the appendix for greater clarity.
>
> ---
>
> *Reference:*
>
> *[1] Wu, Zhenbang, et al. "An iterative self-learning framework for medical domain generalization." NeurIPS 2023.*
>
> *[2] Yang, Chaoqi, M. et al. "ManyDG: Many-domain Generalization for Healthcare Applications." ICLR 2023.*
>
> *[3] Yang, Chaoqi, et al. "Pyhealth: A deep learning toolkit for healthcare applications." ACM SIGKDD  2023.*

---

> ### Author Response · Authors · 2025-12-02
>
> Even though we understand that you cannot reply to our response, we sincerely appreciate your professional and insightful comments (especially **W1-4** and **Q2-3**) which help us strengthen the narrative and more clearly articulate our motivation and contributions.
>
> ---
>
> Since we have also uploaded the revised PDF for the rebuttal, we also provide below a concise guide to help you locate our latest updates related to your suggestions:
>
> - **W1 [Section 1, Line 063–072, Page 2]:** We added an additional paragraph describing the non-trivial gap between using ontologies and addressing DG problems.
>
> - **W2 [Appendix K, Line 1160–1200, Page 22]:** We extended the conclusion to highlight our framework’s potential for multi-modal clinical data and generalization beyond healthcare.
>
> - **W3 [Appendix F.3, Line 1002–1025, Page 19]:** We evaluated cross-institutional generalization on the eICU dataset.
>
> - **W4 [Section 4.4, Line 457–485, Page 9]:** We added a case study analyzing and interpreting the discovered domains after hierarchy pruning.
>
> - **W5:**
>
>   - Abstract: Line 015-017 & 020-021, Page 1
>
>   - Preliminaries: Section 2, Line 099-103, Page 2
>
>   - Methodology: Blue text on Page 3-4
>
>   - Notation: Appendix C, Line 864-900, Page 17
>
> - **Q1:** See Appendix J, Line 1134-1160, Page 22
> - **Q2:** See Table 3, Line 383-385, Page 8
> - **Q3:** See Appedix L, Line 1203-1220, Page 23
>
> We hope these clarifications address your concerns.

---

### Official Review · Reviewer_gkNx · 2025-10-31

**Soundness:** 2
**Presentation:** 3
**Contribution:** 3
**Rating:** 6
**Confidence:** 3

**Summary:**

This paper tackles the critical problem of domain generalization (DG) in clinical prediction, where models trained on Electronic Health Records (EHR) from one population or time period fail when applied to another. The authors argue that standard DG methods are ill-suited for healthcare because EHR data lacks the explicit domain labels found in other fields, and methods that simply cluster patient features ignore the rich clinical semantics that define patient cohorts. To solve this, the paper proposes UDONCARE, a novel framework that, for the first time, uses medical ontologies to actively discover latent, clinically meaningful domains. The framework operates in two main steps: first, a hierarchy-pruning algorithm analyzes the disease hierarchy to identify an optimal level of abstraction, merging overly specific leaf nodes (e.g., "congestive heart failure") into more general, robust ancestor nodes (e.g., "cardiovascular disease") based on node scores and a beam search. Overall result are satisfied.

**Strengths:**

1. The paper's core strength is its novel use of medical ontologies to solve a DG problem. Instead of treating domains as arbitrary, data-driven clusters, it grounds them in established medical knowledge. This makes the discovered domains more interpretable and robust.
2. The hierarchy-pruning algorithm is a non-trivial and well-designed contribution. It correctly identifies that neither leaf-level codes nor root-level categories are optimal, and it provides a principled method to find the best-fitting level of abstraction. The ablation study in Figure 3 confirms this custom pruning method is superior to standard k-Means, hierarchical clustering, or simpler tree pruning.
3. UDONCARE shows consistent and often substantial performance gains over a wide range of baselines on four distinct prediction tasks and two large-scale public datasets.

**Weaknesses:**

1. The framework's success seems to hinge almost entirely on the ICD-9-CM disease hierarchy. The authors note in the appendix that adding procedure and medication codes yielded "marginal" and "inconsistent" gains. This is a limitation, as it suggests the model may not handle domain shifts caused by factors not captured in the disease hierarchy.
2. The study uses a temporal split (e.g., pre-2017 vs. post-2017) as its proxy for domain shift. While this is a valid and common setup, it is not the only (or even most difficult) type of shift. The paper's introduction mentions generalizing across different hospitals, but this cross-institutional generalization is not tested.
3. The domain discovery algorithm is multi-staged and complex, involving hierarchy-aware node initialization, a specific scoring function, a bottom-up pass, and a rectified beam search. This could create a higher barrier for reproducibility and adoption compared to simpler methods.

**Questions:**

1. Your domain discovery relies almost exclusively on the ICD-9 disease hierarchy, and you found that adding other codes didn't help. How would UDONCARE handle a domain shift that is uncorrelated with disease codes, such as a change in hospital billing practices, documentation standards, or the introduction of a new EHR system?
2. The pruning algorithm is designed to find an optimal set of ancestor nodes. What did the result look like in practice? Did the model converge on very high-level concepts, mid-level ones, or a mix? How interpretable were these auto-discovered domains?

---

> ### Author Response · Authors · 2025-11-21
> **Official Comment by Authors (Part 1/2)**
>
> We appreciate your positive feedback and your recognition of the contributions.
>
> **W1: May not handle domain shifts caused by factors not captured in the disease hierarchy.**
>
> Our reliance on the ICD-9-CM is not due to a limitation of our model, but because diseases are the most direct and required source of patient information. Almost all EHR prediction tasks rely on diagnostic codes as a primary input, and diseases exert strong influence on domain formation. Hence, the disease ontology serves as the most appropriate basis in the general clinical setting. It is also important to note that UdonCare is not restricted to disease ontologies. For instance, using a medication ontology (ATC classification) might be equally or even more appropriate in drug recommendation. Our model directly supports such substitution. In this paper, we want to align the used ontology across four tasks in our experiments.
>
> The "marginal" and "inconsistent" gains mentioned in the appendix refer to the case where multiple ontologies are used simultaneously (e.g., disease + drug). In this setting, we might get fine-grained domains: patients with the same disease may be split into different domains solely due to distinct medication regimens, which can weaken the benefits of ontology-guided grouping. Therefore, our conclusion is simply that using several ontologies at the same time is less efficient. Instead, we encourage practitioners to select the most appropriate ontology according to the task. We will revise the manuscript to make this point clearer. Thank you for highlighting this concern!
>
> **W2: The cross-institutional generalization is not tested**
>
> Following the setting used in SLDG [1], we extend our evaluation to the eICU dataset, where hospitals can be categorized into four groups based on their locations (Midwest, Northeast, West, and South). To quickly deploy our model on this dataset, we directly use the processing procedure of PyHealth [3]. We evaluate the performance of our model under spatial domain shifts by treating the Midwest group as the target domain and the remaining groups as the source domain. We use diagnosis and treatment codes to construct patient records, as medications in eICU are not encoded with standard IDs.
>
> The table below reports the results for readmission and diagnosis prediction. We observe that our model continues to demonstrate the superiority over baselines. We will include these additional results in the appendix of the revised version.
>
> |Models|AUPRC (Readmission)|AUROC (Readmission)|w-F1 (Diagnosis)|R@10 (Diagnosis)|
> |--------|-----------|-----------|-----------|-----------|
> |Oracle|22.74 (0.13)|69.29 (0.13)|63.18 (0.08)|79.05 (0.09)|
> |Base|11.97 (0.08)|51.31 (0.06)|53.89 (0.05)|71.40 (0.05)|
> |DANN|12.75 (0.16)|54.28 (0.14)|58.41 (0.18)|75.66 (0.24)|
> |CondAdv|13.16 (0.18)|53.97 (0.12)|57.04 (0.23)|73.92 (0.17)|
> |MLDG|14.82 (0.11)|54.25 (0.17)|58.13 (0.28)|75.31 (0.21)|
> |IRM|15.73 (0.19)|58.31 (0.21)|59.22 (0.16)|76.46 (0.15)|
> |PCL|15.14 (0.23)|58.06 (0.24)|-|-|
> |ManyDG|17.53 (0.17)|60.20 (0.10)|60.61 (0.38)|76.94 (0.21)|
> |SLDG|16.71 (0.21)|59.54 (0.18)|-|-|
> |**UdonCare**|**18.37 (0.18)**|**62.12 (0.11)**|**61.83 (0.14)**|**77.02 (0.11)**|
>
> **W3: The algorithm is multi-stage and complex for reproducibility and adoption than simpler methods**
>
> We agree that the domain discovery module contains several steps, but complexity of rules doesn't stand for hard adoption. Our model is lightweight in terms of implementation: aside from a single MLP used to obtain domain features, the hierarchy pruning can be viewed as a variant of a standard tree-pruning algorithm, with an adjusted scoring rule tailored to medical ontologies. This module does not introduce additional learnable parameters.
>
> We also note that simpler DG baselines do not imply lower algorithmic or computational complexity: (1) most of them cannot operate without assuming domain IDs [1, 2]; (2) some methods, such as meta learning variants, require assuming a large number of domains and often involve much longer training time [2]. In contrast, our method aims to balance structural clarity with practical utility: the pruning algorithm is deterministic, parameter-free, and designed to leverage the hierarchical structure already present in medical ontologies.

---

> ### Author Response · Authors · 2025-11-21
> **Official Comment by Authors (Part 2/2)**
>
> **Q1: How would UDONCARE handle a domain shift that is uncorrelated with disease codes, such as a change in hospital billing practices, documentation standards, or the introduction of a new EHR system?**
>
> The scenarios mentioned above represent system-level shifts rather than clinical-covariate shifts. These shifts do not alter the underlying clinical content but instead affect how information is recorded or encoded.
>
> In practice, even when an EHR system is updated or replaced, hospitals are still required to follow standardized clinical coding rules. For example, the transition from ICD-9-CM to ICD-11-CM mandates that all institutions map diagnoses to the ICD-11 standard for billing and documentation. Since these transitions are governed by formal guidelines, both public and private EHR systems provide official cross-referencing files, enabling a direct translation between code systems. In such cases, UDONCARE can simply operate on ontologies of new encoding systems (e.g. ICD-11) without loss of functionality.
>
> Moreover, as we noted in our response to Weakness 1, we can use the more appropriate ontology for certain task (e.g. ATC ontology in drug recommendation), allowing us to capture domain shifts beyond disease codes when relevant to the task. We acknowledge that our method depends on the existence of ontologies. However, most EHR datasets are encoded using standardized coding systems, making our method widely applicable in real clinical settings.
>
> **Q2: Results and interpretation of domain separation after pruning hierarchy**
>
> We agree that showing discovered domains would enhance both interpretability and practical value of our model. We will include one more case study to focus on the pruned disease nodes and analyze the resulting latent domains. This qualitative analysis will help clarify the clinical-relevant patterns captured by the discovery process. We appreciate your suggestion, and we believe this addition will strengthen the paper.
>
> ---
>
> *Reference:*
>
> *[1] Wu, Zhenbang, et al. "An iterative self-learning framework for medical domain generalization." NeurIPS 2023.*
>
> *[2] Yang, Chaoqi, M. et al. "ManyDG: Many-domain Generalization for Healthcare Applications." ICLR 2023.*
>
> *[3] Yang, Chaoqi, et al. "Pyhealth: A deep learning toolkit for healthcare applications." ACM SIGKDD  2023.*

---

> ### Comment · Reviewer_gkNx · 2025-11-28
> **Response**
>
> Thanks for your detailed rebuttal. Most of my concerns have resolved. However, I agree with reviewer J8r2's point and I cannot raise my score.

---

> > ### Author Response · Authors · 2025-12-02
> >
> > Thanks for your response and for acknowledging that most of your concerns have been resolved! We truly appreciate your detailed insights and suggestions for our paper, especially **W1** and **Q2**, which further strengthen the work.
> >
> > The score you have given already reflects your recognition of our work, and it is entirely reasonable that you may still have reservations about whether it deserves a higher rating. We believe we share the same goal: to make this work as solid and comprehensive as possible.
> >
> > Regarding reviewer J8r2’s point, we would appreciate it if you could check our latest response to them. In short, we have addressed their most critical concern and received their acknowledgment, and we commit to releasing the fully tested and complete code upon acceptance.
> >
> > ---
> >
> > Since we have also uploaded the revised PDF for the rebuttal, we also provide below a concise guide to help you locate our latest updates related to your suggestions:
> >
> > - **W1 [Appendix G, Line 1049–1061, Page 20]:** We added an explanation of the results and clarified our motivation for using only condition ontologies.
> > - **W2 [Appendix F.3, Line 1002–1025, Page 19]:** We tested cross-institutional generalization on the eICU dataset.
> > - **Q2 [Section 4.4, Line 457–485, Page 9]:** We added a case study analyzing domain separation after pruning the hierarchy.

---

### Official Review · Reviewer_7SsM · 2025-10-31

**Soundness:** 3
**Presentation:** 3
**Contribution:** 3
**Rating:** 6
**Confidence:** 2

**Summary:**

In this paper, the authors propose UDONCARE, a framework that leverages medical ontologies to enhance domain generalization in electronic health record (EHR) prediction. Instead of relying on predefined domain labels, UDONCARE automatically discovers clinically meaningful latent domains by pruning hierarchies such as ICD-9 and learning domain-invariant features through mutual learning. Evaluated on MIMIC-III and MIMIC-IV across multiple prediction tasks, the model achieves strong generalization under domain shifts while maintaining computational efficiency. The work provides a novel, knowledge-guided approach for robust healthcare prediction across unseen patient populations.

**Strengths:**

1. The motivation of this study has been clearly illustrated. Benefitting from that, it becomes reasonable for readers to comprehend the necessity of using both Step 1 and 2 to tackle the problem accordingly.

2. Empirical coverage is broad: four clinically meaningful tasks, two large public datasets, and eight competitive baselines, demonstrating general usability as well as advanced performance,e.g., sizeable AUPRC gains on mortality and readmission have beewhile maintaining computation comparable to prior work.

3. Comprehensive analysis, ablation studies and side studies such as runtime test provide valuable insights regarding characteristics of UDONCARE.

**Weaknesses:**

1. There are still some hyperparameter studies that could have been done for a better understanding of UDONCARE. For instance, the updating frequency of M (mentioned in Line 269) can be further investigated.

2. Some technical details can be further elaborated. For instance, how the loss terms in Eq. 9 functions together during the optimization? Per description, I am assuming they are added altogether without any weighting, but please explicitly detail it in Eq. 9.

**Questions:**

I am personally wondering whether this framework can be expanded to general domain, where features are also represented within an ontology, e.g., e-commerce.

---

> ### Author Response · Authors · 2025-11-21
>
> We appreciate your positive feedback and your recognition of the contributions.
>
> **W1:  The frequency of iterative training could be investigated**
>
> Your insight regarding the importance of studying the update frequency aligns well with our design motivation. In fact, we have already conducted the corresponding analysis in **Section 4.3**, where **Figure 3 (left)** examines the effect of the number of iterations, which directly corresponds to the update frequency of M. As shown in Figure 3, updating M every 20 epochs for a total of 3 iterations achieves the best balance between performance and computational cost.
>
> **W2: Are the loss terms in Eq. 9 functions added together without any weighting? Some technical details can be further elaborated.**
>
> About question in Eq. (9), both the KL-divergence term and the supervised cross-entropy term use the same tradeoff parameter $\lambda$ in the definitions of $\mathcal{L} _ p$ and $\mathcal{L} _ h$. The final training objective is obtained by summing the two components, i.e. $\mathcal{L}=\mathcal{L} _ p+\mathcal{L} _ h$. We appreciate your suggestion to make this explicit.
>
> We will revise the paragraph after Eq. (9) to make this clearer as follows:
>
> > where $D _ {\text{KL}}(\hat{y} _ {*} | \tilde{y})$ denotes the KL Divergence  (Van Erven & Harremos, 2014) sharing the same tradeoff parameter $\lambda$, $\ell(\cdot)$ denotes the binary cross-entropy, and $\tilde{y}$ is the average probability of $\hat{y} _ p$ and $\hat{y} _ h$. These two losses are calculated jointly $\mathcal{L} _ p+\mathcal{L} _ h$ to let $d _ \eta$ and $q _ \xi$ regularize one another, stabilizing the learning of $q _ \xi$ with less parameters. Following the domain generalization setting, we adopt $\hat{y}_h$ as the final prediction.
>
> Furthermore, we will add additional clarifications and illustrations in the methodology section. Please also refer to our response to Reviewer eoYo for further elaboration.
>
> **Q1: Whether it can be expanded to general domain**
>
> Thanks for your perspective on the broader applicability of ontology-guided domain discovery. The core ideas behind UdonCare are not limited to the healthcare setting. Any domain where features are organized within a structured ontology could benefit from similar mechanisms for domain discovery and invariant learning. To maintain a clear scientific scope, we did not extend our model to other domains. Nevertheless, we are excited about the possibility that future works may build upon our model in other ontology-rich domains such as e-commerce or large-scale knowledge-driven applications.

---

> ### Author Response · Authors · 2025-12-02
>
> Even though we understand that you cannot reply to our response, we sincerely appreciate your valuable feedback (especially **W1** and **Q1**) which has provided us with meaningful opportunities to further highlight the potential of our work.
>
> ---
>
> As we have uploaded the revised PDF for the rebuttal period, we provide below a concise guide to help you locate the corresponding updates:
>
> - **W1 [Section 4.3, Line 400–421, Page 8]:** The existing Figure 3 (left) already examines the effect of the number of iterations, which directly reflects the update frequency of M.
> - **W2 [Section 3.3, Line 289–293, Page 6]:** We improved the paragraph following Eq. (9) to enhance clarity.
> - **Q1 [Appendix K, Line 1160–1200, Page 22]:** We extended the discussion in our conclusion to illustrate the potential of our framework for multi-modal clinical data and for generalization beyond healthcare.
>
> We hope these clarifications address your concerns.

---

### Official Review · Reviewer_eoYo · 2025-11-03

**Soundness:** 3
**Presentation:** 2
**Contribution:** 3
**Rating:** 4
**Confidence:** 4

**Summary:**

This paper aims to enhance the performance of predictive healthcare using EHR data by considering domain generalization. Specifically, this work iteratively divides patients into latent domains guided by the medical code (e.g., ICD) hierarchy. Within the hierarchy, each leaf node (i.e., code) can be represented by the embedding either from the model or from the entity name embedding. Then, all other nodes' embedding can be obtained using all its descendants. After that, each node can be assigned a score about coverage, purity, and depth borrowing the idea of information gain. Finally, if a parent node's score is higher than all its children, all children will be removed and if it is lower than all its children, the parent will be removed. Otherwise, the Beam-Search algorithm will be applied to either include or remove the parent using the Silhouette score. Now, given a pruned set of codes, patients can be divided into latent domains using the domain labels under those pruned codes and the component invariant to domain shifts can be obtained for training together with the original prediction loss. The experiments comparing with DG baselines using both MIMIC III and MIMIC IV on four different tasks show consistent improvement of performance.

**Strengths:**

1) Domain shift is a challenging concern within healthcare and the idea of using medical code hierarchy to guide the patient partition is interesting and sound.

2) The proposed pruning algorithm to include either the parent only or the children only codes in the hierarchy can not only improve the efficiency but also cover the necessary information hidden in the hierarchy.

3) Consistent performance improvement can be observed over multiple healthcare predictive tasks using both MIMIC III and MIMIC IV compared to a comprehensive list of baselines.

**Weaknesses:**

1) Some technical discussions or explanations are not clear, which makes it hard to follow and verify. For example, the step on the lowest common ancestor was to increase the similarity between certain node pairs (e.g., sharing the same parent). Right? It is not clear how the most similar pair was determined.

2) In the step of hierarchy pruning, some details are missing. For example, we may have a case that in the lowest level, a parent (e.g., node A) has been removed due to the lower score compared to all it children in the leaves. However, in the next level, the parent's parent (e.g., B is a parent of A) is also having a lower score than all it children. Then, we need to keep A and remove B. How this conflict has been resolved?

3) In the step of domain searching, for each parent and children pair, are we also either including the parent only or the children only? The score is evaluated by checking if the node is pruned, how the separation is done. Only parent nodes are considered for this separation? It is not clear. What kind of features have been used to calculate the Silhouette score?

4) For the learning objective, it seems that two components in (9) are combined together. Is it directly summing them with the KL divergence sharing the same tradeoff parameter? This is also not clear.

**Questions:**

Questions can be found in the above weakness section.

---

> ### Author Response · Authors · 2025-11-21
>
> Thank you for your insightful and constructive comments. We have carefully considered all the feedback and provided point-to-point responses below.
>
> **W1: Purpose of updating the lowest common ancestor (LCA) and how the most similar pair is determined**
>
> In this stage, our goal is to make node features involve hierarchical information on the ontology. When two nodes exhibit the high cosine similarity, their LCA is the earliest shared point that can reflect such similarity, and then this ancestor is more likely to be a meaningful separation point during domain discovery.
>
> Concretely, we compute the pairwise **cosine similarities** among all current nodes, rank these pairs from highest to lowest similarity, and iteratively update the LCA embeddings in this order. For each iteration, we take the pair with the current highest similarity and update their LCA as described. This process continues sequentially down the similarity list until the remaining highest similarity falls below the threshold $ρ = 0.3$.
>
> We will clarify this stage more explicitly with equations and illustrations in Appendix.
>
> **W2: Conflicts among child nodes ({C}), parent node (A), and parent's parent node (B)**
>
> We appreciate you bringing this potential conflict to our attention. After careful consideration, we believe there is no conflict for the following reasons:
>
> [Your example] When S(A) < min({C}) and S(B) < S(A);
>
> Results: A and B will be removed, and {C} will be involved in the candidate list;
>
> Explanation: when A is removed, {C} will be the new children of B, so that B will be then removed;
>
> I think the main reason why you have this concern is that we lack present the discipline that
>
> 1. {C} will be the new children of B after dropping A.
> 2. If S(B) has even larger value than the maximum score among all {C} and other (updated) children under B, B will be the latest candidate for pruning.
>
> We will reorganize and clarify these rules more explicitly in the methodology.
>
> **W3: Details about Hierarchy Pruning**
>
> We adopt a beam search strategy that expands decisions pair by pair while retaining only the top-k candidate pruning paths. The Silhouette score is used to evaluate the overall pruning configuration, not individual parent nodes. For each candidate generated during beam search, we construct a temporary pruned node set composed of decisions made so far; flagged pairs that have not yet been decided default to retaining their parent node to ensure complete coverage. The Silhouette score is then computed on this full hierarchical partition, reflecting the global separation quality of leaf assignments under the current pruning scheme. Concretely, we directly use the leaf node features, which are first reduced to a 4-dimensional space using UMAP. The Silhouette value is then calculated by intra-cluster and nearest-cluster distance based on these reduced features.
>
> We will add these detailed clarifications in the appendix.
>
> **W4: Whether KL and Supervised Terms share the same $\lambda$ parameter**
>
> Thank you for pointing this out. Yes, both the KL-divergence term and the supervised cross-entropy term use the same tradeoff parameter $\lambda$ in the definitions of $\mathcal{L} _ p$ and $\mathcal{L} _ h$. The final training objective is obtained by summing the two components, i.e. $\mathcal{L}=\mathcal{L} _ p+\mathcal{L} _ h$.
>
> We will revise the paragraph after Eq. (9) to make this clearer as follows:
>
> > where $D _ {\text{KL}}(\hat{y} _ {*} | \tilde{y})$ denotes the KL Divergence  (Van Erven & Harremos, 2014) sharing the same tradeoff parameter $\lambda$, $\ell(\cdot)$ denotes the binary cross-entropy, and $\tilde{y}$ is the average probability of $\hat{y} _ p$ and $\hat{y} _ h$. These two losses are calculated jointly $\mathcal{L} _ p+\mathcal{L} _ h$ to let $d _ \eta$ and $q _ \xi$ regularize one another, stabilizing the learning of $q _ \xi$ with less parameters. Following the domain generalization setting, we adopt $\hat{y} _ h$ as the final prediction.
> ---
>
> We are working diligently to improve our paper based on your valuable and insightful comments.

---

> > ### Author Response · Authors · 2025-12-02
> >
> > Even though we understand that you cannot reply to our response, we still want to express our sincere appreciation for your detailed suggestions and insights (especially **W1** and **W3**) regarding the clarity of our methodological descriptions. Your comments have genuinely helped us strengthen the paper.
> >
> > ---
> >
> > As we have uploaded the revised PDF for the rebuttal period, we provide below a concise guide to help you locate the corresponding updates:
> >
> > - **W1 [Section 3.1, Line 174–183, Page 4]:** We revised the description to improve clarity and added an explanation of how the LCA is determined in each iteration.
> >
> > - **W2 [Section 3.1, Line 202–203, Page 4]:** We added a sentence clarifying the inheritance rule between parent and child nodes to avoid ambiguity.
> >
> > - **W3 [Section 3.1, Line 208–215, Page 4]:** We refined the description of the domain-searching step and provided more detail on how decisions are made based on Silhouette Scores.
> >
> > - **W4 [Section 3.3, Line 289–293, Page 6]:** We improved the paragraph following Eq. (9) to enhance clarity.
> >
> > We hope these clarifications address your concerns.

---

### Author Response · Authors · 2025-12-02
**General Response to All Reviewers**

We sincerely thank all reviewers for their thoughtful feedback. Reviewers agreed that our paper makes a clear and meaningful contribution: UdonCare introduces a novel ontology-based pruning algorithm for addressing clinical domain generalization problems (eoYo, 7SsM, gkNx, o6w5), demonstrates strong generalization across multiple EHR predictive tasks and public datasets (eoYo, 7SsM, gkNx, J8r2), and enables efficient iterative updates through hierarchy pruning (eoYo, 7SsM, o6w5). Several reviewers also highlighted that grounding the pruning algorithm in established medical knowledge allows UdonCare to discover robust and interpretable patient domains, rather than arbitrary data-driven clusters (eoYo, 7SsM, gkNx, o6w5). Overall, reviewers found the work well-motivated, thoroughly evaluated, and practically relevant (eoYo, 7SsM, gkNx, o6w5, J8r2). We are deeply grateful for their insights, which have significantly strengthened the quality of this work.

Below is a summary of the key revisions of the revised PDF made in response to the reviewers’ comments:

- **Methodology (eoYo, 7SsM):** We improved the exposition of our pruning algorithm, including clearer explanations of LCA determination, inheritance rules across hierarchy levels, and the beam-search domain selection procedure **(blue text on Page 4 & Appendix I, J, and K)**. We also refined the description following Eq. (9) to enhance conceptual clarity **(Line 290-293 onPage 6)**.
- **Experiments (gkNx, o6w5):** We added a cross-institutional generalization experiment on the eICU dataset **(Section 4.4 on Page 9)** and incorporated a case study analyzing discovered domains after pruning **(Appendix F.3 on Page 19)**. Additional clarification was also provided for Table 3 **(Appendix L on Page 23)**.
- **Introduction & Conclusion (7SsM, o6w5):** We expanded the introduction to explain the non-trivial gap between ontology usage and DG problems **(Line 063-072 on Page 2)**, extended the conclusion to discuss broader multi-modal and generalization potential beyond healthcare **(Appendix K on Page 22)**.

Sincerely,

The Authors

---

### Meta-Review · Area_Chair_V1tp · 2026-01-09

**Summary:**

This paper proposes UdonCare, a hierarchy-guided framework for unseen domain discovery and domain generalization in predictive healthcare, leveraging medical ontologies to partition patients into latent domains. Reviewers generally agreed that the problem is important and that grounding domain discovery in clinical knowledge is well motivated. The proposed hierarchy pruning mechanism and empirical evaluation across multiple tasks and datasets demonstrate promising results. However, despite these strengths, reviewers raised concerns about methodological clarity, robustness, evaluation design, and overall contribution strength. After considering the rebuttal and discussion, the paper does not meet the acceptance bar.

**Reviewer Concerns:**

Several reviewers noted that while the idea of ontology-guided domain discovery is appealing, the methodology is complex and insufficiently clear, with key components (e.g., hierarchy pruning rules, scoring functions, and optimization objectives) requiring substantial clarification. Although the authors addressed many of these issues in the rebuttal, the approach remains difficult to verify and reproduce.

Concerns were also raised about the evaluation setup, including the reliance on specific temporal splits, limited diversity of domain shift scenarios, and dependence on disease ontologies as the primary driver of domain discovery. While additional experiments (e.g., cross-institutional evaluation on eICU) were added, reviewers remained unconvinced that the empirical evidence fully supports the generality of the claims. The baseline selection and experimental design were debated extensively, and these issues were not resolved to a level that established clear consensus.

Overall, reviewers viewed the contribution as incremental, with limited technical depth beyond existing DG and ontology-based modeling approaches.

**Reviewer Scores:**

Reviewer scores were mixed but leaned negative, with one clear reject and several marginal scores indicating that reviewers “would not mind” either outcome. The overall consensus supports rejection.

---

### Decision · Program_Chairs · 2026-01-26

Reject